



# Validation of MAX-DOAS retrievals of aerosol extinction, SO$_2$ and NO$_2$ through comparison with lidar, sun photometer, Active-DOAS and aircraft measurements in the Athabasca Oil Sands Region.

Zoë Y. W. Davis[1], Udo Frieβ[2], Kevin B. Strawbridge[3], Monika Aggarwaal[1], Sabour Baray[4], Elijah G.
Schnitzler[5], Akshay Lobo4[4,6], Vitali E. Fioletov[3], Ihab Abboud[3], Chris A. McLinden[3], Jim Whiteway[1],
Megan D. Willis[5,7], Alex K. Y. Lee[8], Jeff Brook[3,9], Jason Olfert[10], Jason O'Brien[3], Ralf Staebler[3], Hans
D. Osthoff[11], Cristian Mihele[3], and Robert McLaren[4].

[1] Department of Earth and Space, York University, Toronto, M3J 1P3, Canada
[2] Institute of Environmental Physics, Heidelberg, 69120, Germany
[3] Environment and Climate Change Canada, Toronto, M3H 5T4, Canada
[4] Centre for Atmospheric Chemistry, York University, Toronto, M3J 1P3, Canada
[5] Department of Chemistry, University of Toronto, M5S 3H6, Canada
[6] *now at* Department of Orthopaedics, University of British Columbia, Vancouver, V5Z 1M9, Canada
[7] *now at* Chemical Sciences Division, Lawrence Berkeley National Lab, Berkeley, California, 94720, USA
[8] Department of Civil and Environmental Engineering, National University of Singapore, 117576, Singapore
[9] *now at* Dalla Lana School of Public Health, University of Toronto, M5S 3H6, Canada
[10] Department of Mechanical Engineering, University of Alberta, Edmonton, Alberta, T6G 1H9
[11] Department of Chemistry, University of Calgary, Calgary, T2N 1N4, Canada

*Correspondence to*: Zoë Davis (zoeywd@yorku.ca)

**Abstract.** Vertical profiles of aerosols, NO$_2$, and SO$_2$ were retrieved from Multi-Axis Differential Optical Absorption Spectroscopy (MAX-DOAS) measurements at a field site in northern Alberta, Canada, during August and September 2013. The site is approximately 16 km north of two mining operations that are major sources of industrial pollution in the Athabasca Oil Sands Region. Pollution conditions during the study ranged from atmospheric background conditions to heavily polluted with elevated plumes, according to meteorology. This study aimed to evaluate the performance of the aerosol and trace gas retrievals through comparison with data from a suite of other instruments. Comparisons of AODs from MAX-DOAS aerosol retrievals, lidar vertical profiles of aerosol extinction, and AERONET sun photometer indicate good performance by the MAX-DOAS retrievals. These comparisons and modelling of the lidar S-ratio highlight the need for accurate knowledge of the temporal variation in the S-ratio when comparing MAX-DOAS and lidar data. Comparisons of MAX-DOAS NO$_2$ and SO$_2$ retrievals to Pandora spectral sun photometer VCDs and Active-DOAS mixing ratios indicate good performance of the retrievals except when vertical profiles of pollutants within the boundary layer varied rapidly, temporally and spatially. Near-surface retrievals tended to overestimate Active-DOAS mixing ratios. The MAX-DOAS observed elevated pollution plumes not observed by the Active-DOAS, highlighting one of the instrument's main advantages. Aircraft measurements of SO$_2$ were used to validate retrieved vertical profiles of SO$_2$. Advantages of the MAX-DOAS instrument include increasing sensitivity towards the surface and the ability to simultaneously retrieve vertical



profiles of aerosols and trace gases without requiring additional parameters such as the S-ratio. This complex dataset provided a rare opportunity to evaluate the performance of the MAX-DOAS retrievals under varying atmospheric conditions.

## 1 Introduction

The Athabasca Oil sands operations in Alberta contain significant sources of industrial atmospheric pollutants such as
sulphur dioxide ($SO_2$) and nitrogen dioxide ($NO_2$) (Government of Canada, 2018c, 2018d). Oil extraction and upgrading activities such as surface mining, acid gas flaring, and transporting materials in heavy hauler trucks emit aerosols and trace gas pollutants (Liggio et al., 2016). Pollutant emissions from the industrial smokestacks result in uplifted profiles with the potential to be transported farther downwind compared to emission released at the surface, particularly for stacks with high volume flow rates and temperatures that can rise high in the atmosphere (Zhang et al., 2018). While the Athabasca Oil Sands
Region (AOSR) experiences moderate annual average concentrations of $SO_2$ relative to all Canadian in-situ stations, the short-term concentrations can be significantly higher than in most Canadian cities (Government of Canada, 2018a). The AOSR contains some of the few monitoring sites in Canada that experience peak 1-hour average concentrations of $SO_2$ of greater than 70 ppb (Government of Canada, 2018a), which is the new 2020 Canadian Ambient Air Quality Standard for $SO_2$ (Canadian Council of Ministers of the Environment, 2014). $SO_2$ concentrations of up to 131 ppb were also observed by
aircraft measurements downwind of an AOSR industrial facility in 2013, approximately midway between Syncrude Mildred Lake Plant and Fort McKay (Baray et al., 2018). High concentrations of $SO_2$ over short durations are a health concern because negative pulmonary and respiratory effects of inhalation can occur after exposure periods as small as 10 minutes (Health Canada, 2016; World Health Organisation, 2006). Exposure to $NO_2$ at high concentrations over short-term is also associated with significant health impacts (World Health Organisation, 2006) and $NO_x$ (NO + $NO_2$) is a precursor to
tropospheric ozone ($O_3$), acid rain and fine particulate matter (Seinfeld and Pandis, 2006).

Emissions of $NO_x$ and $SO_2$ lead to the formation of nitrate and sulphate aerosols, which constitute a significant fraction of the $PM_{2.5}$ air mass in urban and industrially-impacted regions (Pui et al., 2014). The highest peak and annual average $PM_{2.5}$ concentrations in Canada in 2016 were observed at two monitoring stations within Fort McMurray with annual averages of over 18 µg m$^{-3}$ compared to 8 µg m$^{-3}$ in an industrial area of Toronto, Ontario (Government of Canada, 2018a). Exposure to
$PM_{2.5}$ leads to adverse effects on respiratory and cardiovascular systems (World Health Organisation, 2006).

In the troposphere, nearly all $SO_2$ is oxidized to $H_2SO_4$ aerosol through reactions in the gas and aqueous phases. The hydroxyl (OH) radical initiates the oxidation route of $SO_2$ in the gas phase, forming $HOSO_2$ (Holloway and Wayne, 2010). Sulfuric acid ($H_2SO_4$) is formed through further oxidation of $HOSO_2$ and condenses onto already present aerosols or can nucleate with water vapour ($H_2O$) and gaseous ammonia ($NH_3$), forming sulphate aerosol (Kulmala et al., 2004). Aqueous
phase reactions form sulphate aerosol efficiently with $H_2O_2$ and $O_3$ acting as oxidants (Seinfeld and Pandis, 2006). Wet deposition dominates the removal of sulphate aerosol. Therefore, elevated levels of $SO_2$ and $NO_2$ observed over the AOSR region are an environmental concern since atmospheric depositions of sulphur oxides ($SO_x$) and nitrogen oxides ($NO_x$) can



lead to freshwater and soil acidification (Psenner, 1994; Zhao et al., 2009). Deposition of nitrogen compound can harm sensitive ecosystems through eutrophication (excessive nutrient richness) of water bodies (Fenn et al., 2015).

High concentrations of $SO_2$ and other pollutants over the AOSR have prompted measurements using aircraft studies (Baray et al., 2018; Gordon et al., 2015; Liggio et al., 2016, 2019; Simpson et al., 2010), in-situ measurements (Amiri et al., 2018;

Hsu, 2013; Tokarek et al., 2018), sun photometer (Fioletov et al., 2016), and satellite (McLinden et al., 2012, 2014, 2016). Long-term monitoring through satellite measurements is an attractive choice due to the large scale of the operations. However, surface concentrations are difficult to determine accurately from satellite measurements (Fioletov et al., 2016), and data acquisition is limited to the satellite overpass times. Satellite retrievals in the AOSR region are also complicated by multiple factors: landscapes are complex, emissions can change relatively rapidly, and the winds within the higher boundary

layer can quickly disperse pollution emissions. Rapid industrial expansion can also require updating retrieval algorithms (McLinden et al., 2014). Apparent peak concentrations are reduced, and small-scale variability cannot be resolved, due to spatial averaging within the footprint of a pixel that can be large relative to the scale of point-source plumes.

$SO_2$, $NO_2$ and aerosol levels in the total column and near-surface can be simultaneously monitored using the Multi-Axis Differential Optical Absorption Spectroscopy (MAX-DOAS) technique (Honninger et al., 2004). The elevated levels of $SO_2$

observed in the AOSR increase the ease of MAX-DOAS measurements compared to within most Canadian cities, where $SO_2$ levels are significantly lower. Differential Optical Absorption Spectroscopy (DOAS) is a remote sensing technique that quantifies tropospheric trace gases using light spectra and the unique spectral absorption cross sections of trace gases. Since its introduction by Platt et al. (1979) DOAS has been used to quantify trace gases in the troposphere, including $NO_2$, $SO_2$, OH, BrO, $NO_3$, $NH_3$, ClO and others. The technique has the advantage of allowing the simultaneous quantification of

multiple trace gases Platt et al. (2008). The MAX-DOAS method measures scattered sunlight spectra at multiple viewing directions and/or elevation angles to allow sensitive quantification of tropospheric pollutants. Spectra measured at elevation angles close to horizon-pointing have a higher sensitivity to ground-level pollutants since the light paths are longer near the surface (Honninger et al., 2004). Ground-based MAX-DOAS measurements determine tropospheric vertical column densities (VCDs) of trace gases, quantifying total boundary layer pollution loading. VCDs have the advantage of being

independent of boundary layer height and are spatially averaged (horizontally) on the order of a few kilometres along the light path.

Ground-based MAX-DOAS data combined with radiative transfer modelling allows retrieval of vertical profiles of aerosol extinction and trace gases (Frieß et al., 2006; Honninger et al., 2004; Honninger and Platt, 2002; Irie et al., 2008; Wagner et al., 2004). The MAX-DOAS technique has been used to retrieve vertical profiles of aerosol extinction (Clemer et al., 2010;

Frieß et al., 2011; Irie et al., 2008, 2015; Li et al., 2010; Zieger et al., 2011), BrO (Frieß et al., 2011; Honninger and Platt, 2002), HCHO (Heckel et al., 2005; Wagner et al., 2011), $SO_2$ (Tan et al., 2018) and $NO_2$ (Tan et al., 2018; Wagner et al., 2011).

There are few comparisons of vertical profiles of aerosol extinction from MAX-DOAS to vertical profiles from other instruments in the literature. MAX-DOAS aerosol extinction profiles have been compared to smoothed extinction profiles





from a sun photometer (Frieß et al., 2011) and aircraft aerosol profiles (Wagner et al., 2011). Near-surface MAX-DOAS retrievals of aerosol extinction have been compared with in-situ measurements of aerosols (Zieger et al., 2011). There are also relatively few published comparisons of MAX-DOAS AODs with lidar AODs (Irie et al., 2008, 2015). Relatively few studies have focused on MAX-DOAS measurements of anthropogenic $SO_2$ (Irie et al., 2011; Jin et al., 2016; Wang et al.,

2014, 2017; Wu et al., 2018, 2013). Most studies that present MAX-DOAS vertical profile retrievals compare them to trace gas VCDs or near-surface measurements from in-situ or LP-DOAS instruments. Tan et al. (2018) and Wang et al. (2017) compared MAX-DOAS $SO_2$ VCDs to satellite VCDs of trace gases. Tan et al. (2018) and Wagner et al. (2011) compared MAX-DOAS retrievals of vertical profiles of $NO_2$ to satellite VCDs and near-surface $NO_2$ mixing ratios from LP-DOAS, respectively.

In this study, a MAX-DOAS instrument was deployed during a comprehensive air quality campaign conducted during August and September 2013. Pollution conditions ranged from background to heavily polluted with a well-mixed boundary layer to distinctly elevated pollution plumes. Vertical profiles of aerosols, $NO_2$, and $SO_2$ in the troposphere were retrieved using optimal estimation inverse modelling from the MAX-DOAS measurements. These retrievals allowed characterization of the vertical structure of the boundary layer. The retrieval used a two-step approach: 1) aerosol extinction profiles are

retrieved from measured MAX-DOAS $O_4$ Differential Slant Column Densities (dSCDs), and 2) the aerosol extinction profiles are used as forward model parameters for retrieval of trace gas profiles from measured trace gas dSCDs.

Our study adds to the current literature by comparing MAX-DOAS aerosol and trace gas retrievals with data from numerous other instruments deployed during the campaign. The aerosol retrievals were compared to aerosol extinction data from a co-located lidar instrument and a nearby sun photometer. Validation of the aerosol retrievals is essential because these profiles

are used as model parameters for the trace gas retrievals. MAX-DOAS $NO_2$ and $SO_2$ retrievals were compared to mixing ratios from a co-located active-DOAS instrument and tropospheric VCDs of trace gas from a Pandora sun photometer. In-situ measurements of $SO_2$ from an aircraft allowed comparison of MAX-DOAS vertical profiles of $SO_2$. Evaluation of the retrievals was aided by co-located, near-surface measurements of particle size distribution and composition, and nearby, high-resolution measurements of vertical profiles of wind speed and -direction.

The objectives of our study were to 1) determine the factors required to validate MAX-DOAS aerosol retrievals through comparison with lidar and sun photometer data, 2) evaluate the performance of the aerosol and trace gas retrievals through comparison to other datasets, 3) identify conditions that limit the use of the MAX-DOAS technique, and 4) identify conditions under which the MAX-DOAS method was advantageous over other instruments.

This complex dataset from comprehensive measurements in the vicinity of oil sand operations provided a unique opportunity
to test the performance of the MAX-DOAS aerosol and trace gas retrievals.



## 2 Experimental

### 2.1 Field Sites

The MAX-DOAS instrument (Hoffmann Messtechnik GmbH) measured scattered sunlight at an elevation of ~10 m above the surface from Aug. 14 – Sept. 9, 2013 at the Fort McKay South field site (57.149N, 111.642W) north of Fort McMurray,

Alberta concurrent with an Environment and Climate Change Canada (ECCC) intensive measurement campaign (Figs. 1 & S1). A second site was located 4km north of Fort McKay South (Oski-Ôtin; 57.184N, 111.640W) in the Fort MacKay community. Two major sources of aerosols, $NO_2$, $SO_2$ and other pollutants are located south of Fort McKay South: the Syncrude Mildred Lake Plant and the Suncor Millennium Plant, 12 km South and 20 km South-South-East, respectively (Fig. 1). The 2013 NPRI reported emissions of $SO_2$ and $NO_x$ from these facilities were 63 and 14 kilotonnes (kt) and 14 and

8 kt, respectively (Government of Canada, 2018b). Relatively smaller sources of pollutants are located north of Fort McKay South: Shell Jackpine and Muskeg River Mines, CNRL Horizon, and Imperial Oil Kearl Mine (Fig. 1). Tables S1 and S2 show the 2013 NPRI emissions of $SO_2$ and $NO_x$ from these five facilities. A recent study suggests that total industrial emissions of $NO_x$ were underestimated in the NPRI report, particularly for ground sources (Zhang et al., 2018). Since there are $NO_x$ sources that are not included in the NPRI emissions data, also included in Tables S1 and S2 are the 2010 vehicular

emissions associated with each facility and 2012-2013 annual stack and area source emissions from Zhang et al. (2018).

### 2.2 Instrumentation

A mini-MAX-DOAS instrument measured scattered sunlight with a viewing azimuth angle of $155^o$ South-South-East (SSE) at sequential viewing elevation angles $2^o$, $4^o$, $8^o$, $15^o$, $30^o$ and $90^o$ (zenith) above the horizon. The instrument consisted of a

sealed metal box containing entrance optics, UV fibre-coupled spectrograph and all electronics. The instrument field of view was approximately $0.6^o$. Incident light was focused on a cylindrical quartz lens (focal length = 40 mm) into a quartz fibre that transmitted the light into the OceanOptics USB2000 spectrograph. The spectrograph detector was a Sony ILX511 linear silicon Charge-Coupled Device (CCD) array (2048 pixels, pixel size 14x200 microns, signal-to-noise ratio at full signal 250:1). The spectrograph had a spectral range of 290-433 nm, a 50 µm wide entrance slit and a spectral resolution of ~0.6

nm FWHM. The spectrograph was cooled by a Peltier stage to maintain the selected temperature ($5^{oC}$). Spectrometer data was transferred to a laptop computer via USB cable. The instrument was controlled using the software package DOASIS, which allowed automated measurements by JScript programs. The instrument was mounted on an elevated scaffold approximately 10 m above ground level (a.g.l.), approximately at the height of the surrounding forest canopy. Each recorded measurement spectrum was an average of 2000 measured spectra with an exposure time that varied between 50 and 200

milliseconds, depending on the ambient light levels.

MAX-DOAS aerosol and trace gas retrieved data were inter-compared with data from various other instruments deployed during the campaign. Table 1 provides information on these instruments and papers that describe their operation.



An Active-DOAS instrument located at the same site was used to retrieve mixing ratios of $NO_2$ and $SO_2$ at 3.5 m a.g.l. Measurements of trace gases with the active-DOAS system have been described previously (McLaren et al., 2010, 2012; Wojtal et al., 2011) although details changed in the current study. DOAS measurements were made using a modified DOAS 2000 Instrument (TEI Inc.) utilizing a 150W high-pressure Xe-arc lamp and a coaxial Cassegrain telescope. The outgoing

beam traversed the atmosphere for 1.15 km (pathlength =2.3 km) at an average height of 3.5 m a.g.l. where it impacted a retroreflector array composed of 30×2" hollow corner cubes mounted on a raiseable tower. The beam traversed through an exploration line cut (5-10 m wide × 2 km) in a mature coniferous forest. Return light was collected with a 2 m × 600 μm UV transparent fibre optic cable and spectrometer (Ocean Optics USB2000, Grating #10, λ=288-492 nm, 1800 lines mm$^{-1}$, 2048 element CCD, 25 μm slit, UV2 upgrade, L2 lens). Integration times of 30-40 ms and 4000 averages gave ≈ 2 min resolution

with detection limits (3σ) of 120 ppt and 170 ppt for $NO_2$ and $SO_2$, respectively. Xenon lamp, Hg calibration, offset and dark noise spectra were collected for spectral fitting with DOASIS software. A small diffuser was installed in the entrance of the fibre to lower atmospheric turbulence noise (Stutz and Platt, 1993) in addition to using an optical fibre bending mode mixer.

A Pandora spectral sun photometer at Oski-Ôtin measured in direct-sun and zenith-sun viewing modes to retrieve total atmospheric column VCDs of $SO_2$ and $NO_2$ with precisions (1σ) of $4.6 \times 10^{15}$ and $0.3 \times 10^{15}$ molecules cm$^{-2}$, respectively

(Fioletov et al., 2016). Tropospheric VCDs of $NO_2$ were determined from the Pandora total column VCDs by subtracting stratospheric VCDs modelled using the PRATMO stratospheric photochemical box model (McLinden, 2000). PRATMO was used as described in Adams et al. (2016) except monthly-mean OSIRIS ozone profiles (Degenstein et al., 2009) and MODIS surface reflectivities (McLinden et al., 2014) were employed. The Pandora $SO_2$ VCDs presented are assumed to be representative of tropospheric $SO_2$ VCDs since stratospheric $SO_2$ was assumed to be negligible. Pandora trace gas and

MAX-DOAS data were both available for inter-comparison for 4 days during the study. $SO_2$ and $NO_2$ mixing ratios were also measured from the air on board a Convair 580 research aircraft (Baray et al., 2018) using Thermo Scientific 43iTLE and 42i-TL analyzers, respectively, between 12 August and 7 September 2013, including a spiral ascent near Fort McKay South ( Sep 03).

Aerosol optical depths (AOD) at 380 nm and 340 nm were obtained from Level 2.0 AERONET data, measured by second

sun photometer at Oski-Ôtin. Aerosol extinction profiles at 532 nm from 0.1-12 km a.g.l. were retrieved using a ground-based, zenith-pointing lidar operated at Fort McKay South (Strawbridge, 2013). In this study, the lidar profiles from 0.1-4 km were considered in order to match the vertical observation extent of the MAX-DOAS. The lidar has the advantage over sun photometer instruments because it can determine the vertical profile of optical extinction rather than just a column-averaged value but has higher uncertainty when the S-ratio is variable (Strawbridge, 2013). Aerosol extinction profiles are

retrieved from the measurements of the laser return signal using a chosen S-ratio value. The S-ratio is the ratio of the volume extinction coefficient to the backscatter coefficient and dictates the signal strength of the received return of the lidar's pulsed laser source (Strawbridge, 2013). Lidar S-ratios are known to be variable but are often estimated given the type of particles expected in an environment (Irie et al., 2015). The S-ratio depends on the shape, size distribution and chemical composition



of the aerosol particles, as well as the relative humidity (Weitkamp, 2005). A constant lidar ratio ("S-ratio") of 25 was used for the lidar retrievals unless otherwise specified. S-ratios were modelled using Mie scattering theory and measurements of surface-level particle composition and size distribution at Fort McKay South for various times during Aug 23 to determine temporal variability in the S-ratio. Source code for the Mie scatting calculations can be found in (Aggarwal et al., 2018).

Ground-level particle composition was measured using an Aerodyne high resolution soot-particle aerosol mass spectrometer (SP-AMS) (Lee et al., 2019). Particle size distributions were measured using Scanning Mobility Particle Sizer (SMPS) ("dry" line mode) and Aerodynamic Particle Sizer (APS) instruments (see supplementary information in (Tokarek et al., 2018) for more details). Particle diameters measured by the SMPS and by the APS were 0.014-0.74 μm and 0.5-19.81 μm, respectively. Data from these instruments were combined to determine particle size distributions from 0.014 to 19.81 μm,

assuming the particles were unit density. Use of "dry" line mode SMPS increased uncertainties in the size distributions because ambient aerosols have more volume than dry aerosols. However, even in the highest relative humidity range, the ambient aerosol had only 30% more volume compared to the dry aerosol which, assuming spherical particles, only results in a maximum increase in particle diameter of 9%. The resulting error is expected to be much smaller than other errors such as converting mobility and aerodynamic diameters to optical diameters.

A radio acoustic meteorological profiler (windRASS, model MFAS, Scintec, Germany) at Oski-Ôtin measured temperature, wind speed and -direction at 10 m intervals from 40 m to up to a maximum altitude of 800 m (Gordon et al., 2017).

## 2.3 MAX-DOAS Data Analysis

### 2.3.1 MAX-DOAS Fitting

Trace gas Differential Slant Column Densities (dSCDs) were obtained using the DOAS technique (Platt et al., 2008) with DOASIS software (Institute of Environmental Physics, Heidelberg, Germany). All spectra were corrected for dark current and electronic offset and wavelength calibrated using a measurement of a Hg lamp. Table 2 shows the wavelength windows and fit components used to retrieve dSCDs of $NO_2$, $SO_2$ and $O_4$. Cross sections were obtained from the MPI-MAINZ UV/VIS Spectral Atlas of Gaseous Molecules of Atmospheric Interest (Keller-Rudek et al., 2013). Examples of spectral

retrievals of the gases are shown in Fig. S2. Each non-zenith measured spectrum was fit against the closest zenith spectrum in time, also known as the Fraunhofer Reference Spectrum (FRS). The statistical error of the $O_4$ dSCDs was $<1.1 \times 10^{42}$ molecules cm$^{-2}$. The $O_4$ error for off-axis measurements relative to the FRS are $<6\%$ for angles below $30^o$ and $<10\%$ for the $30^o$ measurements. The statistical fit errors of the $SO_2$ and $NO_2$ dSCDs were $0.4-1.2 \times 10^{16}$ and $0.4-1.6 \times 10^{15}$ molecules cm$^{-2}$, respectively. Uncertainties in the absorption cross sections result in systematic errors in the retrieved dSCDs. The reported

uncertainty in the $SO_2$ and $NO_2$ absorption cross sections used is approximately 3% (Bogumil et al., 2003). The absolute value of the $O_4$ cross section and its dependence on temperature is uncertain. Some studies suggest that the absolute value of the cross section may be overestimated by up to 25%, requiring the use of a scaling factor (Clemer et al., 2010; Wagner et





al., 2002, 2009). However, Frieß et al. (2011) found that the best results for measured O$_4$ dSCDs and the retrieved vertical profiles of aerosol extinction retrieved from them were achieved without a scaling factor. Irie et al. (2015) found that a scaling factor of 1.25 resulted in an overestimation of near-surface aerosol extinction coefficients (AECs) but also reduced residuals at high viewing elevation angles. A scaling factor was not used for the O$_4$ fitting in this study.

The SO$_2$ fitting range was determined based on an experiment using an SO$_2$ calibration cell from Resonance Ltd. with a slant column density (SCD) of 2.2x10$^{17}$ (+/- 10%) molecules cm$^{-2}$ placed inside the MAX-DOAS telescope. Scattered solar light spectra were recorded around solar noon at multiple viewing elevation angles above the horizon, followed by a 90$^{o}$ measurement without the cell (the FRS). For each of the measured spectra, dSCDs of SO$_2$ were fit in DOASIS by varying the fitting windows in ~0.3 nm increments with a range of lower and upper limits of 303-318 nm 309-340 nm, respectively. The

fit components are the same as in Table 2. See Supplemental Section 2 for details. The NO$_2$ and O$_4$ fitting ranges were from McLaren et al. (2010) and Frieß et al. (2011), respectively.

## 2.3.2 Retrieval of Vertical Profiles from MAX-DOAS dSCDs using Optimal Estimation

Aerosol and trace gas profiles were retrieved using a two-step approach: 1) aerosol extinction profiles were retrieved from measured MAX-DOAS O$_4$ dSCDs and 2) aerosol extinction profiles were used as forward model parameters for retrieval of

NO$_2$ and SO$_2$ profiles from dSCDs of NO$_2$ and SO$_2$, respectively. Vertical profiles were determined from dSCDs using retrieval algorithms based on the (Rodgers, 2000) optimal estimation technique (Frieß et al., 2011, 2016, 2019). Generally, the desired state of the atmosphere ($\mathbf{x}$) can be estimated from remote sensing measurements ($\mathbf{y}$) using a forward model $\mathbf{F}$.

$$\mathbf{y} = \mathbf{F}(\mathbf{x}, \mathbf{b}) + \boldsymbol{\varepsilon} \qquad\qquad (\,1\,)$$

Where $\boldsymbol{\varepsilon}$ is the measurement error and $\mathbf{b}$ is the vector of model parameters that are assumed to be known and not determined by the modelling, such as aerosol microphysical properties. In this study, the SCIATRAN radiative transfer model was used

as the forward model (Rozanov et al., 2005).

The optimal estimation method determined the most probable atmospheric state, $\hat{\mathbf{x}}$, based on a set of measurements, $\mathbf{y}$, and an a-priori state vector $\mathbf{x_a}$. The $\mathbf{x_a}$ was the best guess of the vertical profile to be retrieved. The $\hat{\mathbf{x}}$ was the aerosol extinctions or the trace gas mixing ratios at a series of altitude intervals, for the aerosol retrieval and trace gas retrievals, respectively. The $\mathbf{y}$ was the O$_4$ dSCDs and the trace gas dSCDs measured at different angles, for the aerosol and trace gas retrievals,

respectively. Note that in our retrievals, $\mathbf{y}$ was the dSCDs measured at sequential elevation angles during 20-minute periods before 17:00 local time and during 30-minute periods after 17:00. The wavelengths for the optimal estimation retrievals of O$_{4,}$ NO$_2$ and SO$_2$ were 360.8, 422.5, and 318.0 nm, respectively.

The optimal estimation solution $\hat{\mathbf{x}}$ is the maximum a posteriori (MAP) solution, which selects the most probable state from the set of possible states described by maximizing the probability of $\mathbf{x}$ occurring given the observations $\mathbf{y}$ (Rodgers, 2000).

The MAP solution is found by minimizing the cost function ($\chi^2$).

$$\chi^2 = (\mathbf{y} - \mathbf{F}(\mathbf{x}, \mathbf{b}))^T S_E^{-1} (\mathbf{y} - \mathbf{F}(\mathbf{x}, \mathbf{b})) + (\mathbf{x} - \mathbf{x}_a)^T S_a^{-1} (\mathbf{x} - \mathbf{x}_a) \qquad\qquad (\,2\,)$$





where $\mathbf{S_a}$ and $\mathbf{S_E}$ are the error covariance matrices associated with the a-priori and measurement vectors, respectively (Rodgers, 2000). The retrieval yields important quantities that allow the characterization of the retrieval. These include the weighting function, $\boldsymbol{K} = \frac{\partial \mathbf{F}}{\partial \mathbf{x}}$, which quantifies the sensitivity of the measurement towards the atmospheric state, and the averaging kernel matrix, $\mathbf{A} = \frac{\partial \hat{\mathbf{x}}}{\partial \mathbf{x}}$, which quantifies the vertical resolution of the retrieval. $\mathbf{A}$ describes the sensitivity of the

retrieved profile to changes in the true atmospheric profile. Rows of $\mathbf{A}$ are averaging kernels for each altitude interval in the retrieved profile. The full width half maximum (FWHM) of each kernel gives an estimate of the retrieval's vertical resolution at height z. Each averaging kernel ideally peaks at a magnitude of 1.0 at the height of the kernel. However, the peak value of a kernel is generally less than 1.0 due to finite vertical resolution and may peak at a slightly different height, resulting in the smoothing of the true atmospheric profile into the retrieved profile.

**Aerosol Extinction Retrievals**

Retrieval of aerosol extinction profiles was non-linear since the aerosol extinction affects the radiative transfer in the forward model. The input for the aerosol retrieval was the measurement vector of the $O_4$ dSCDs at different elevation angles and an a-priori state vector that decreased exponentially with altitude with a scale height of 0.6 km and a surface magnitude of $0.1 km^{-1}$. A single a-priori profile choice is preferable for a set of consecutive measurements where information content is

potentially limited since the a-priori will always have some impact on the retrieved profile (Rodgers, 2000). Otherwise, diurnal and day-to-day trends in the retrieved profiles due to real atmospheric changes could be indistinguishable from changes due to a variable a-priori profile.

Our aerosol retrieval used an iterative algorithm based on the Levenberg-Marquart method (Levenberg, 1944; Marquardt, 1963). For aerosol retrievals, the weighting function $\mathbf{K}$ is calculated using the a-priori $\mathbf{x_a}$ and the measurement vector $\mathbf{y}$. The

$\mathbf{K}$ of each retrieval depended on the state vector and changed depending on the determined aerosol extinction profile. The height resolution of the aerosol extinction vertical profile grid was 100 m with a maximum height of 4 km. A detailed description of the aerosol retrieval algorithm can be found in (Frieß et al., 2006).

**Trace gas Retrievals**

The retrievals of $NO_2$ and $SO_2$ vertical profiles were linear because these weak absorbers do not significantly impact the

radiative transfer. The inputs for the $NO_2$ and $SO_2$ retrievals were the measurement vectors of the $NO_2$ and $SO_2$ dSCDs at different elevation angles, respectively, and an a-priori state vector that decreased exponentially (scale height = 0.6 km and surface magnitude = 30 ppb and 10 ppb for $SO_2$ and $NO_2$, respectively).

In this linear case, the forward model is independent of the atmospheric state $\mathbf{x}$, and the weighting function matrix represents the forward model.

$$\mathbf{y} = \mathbf{F}(\mathbf{x}, \mathbf{b}) + \boldsymbol{\varepsilon} = \mathbf{Kx} + \boldsymbol{\varepsilon} \tag{3}$$

In our retrieval, the box-air mass factors (AMF) that are components of $\mathbf{K}$ were modelled using the Monte Carlo radiative transfer model in SCIATRAN (Deutschmann et al., 2011; Frieß et al., 2010). The aerosol profiles retrieved in step 1 were



used to recalculate the Box-AMFs for each trace gas retrieval since the extinction profiles varied. The height resolution of the trace gas vertical profile grid was 100 m with a maximum height of 4 km a.g.l.

**Determination of Retrieval Errors**

The retrieval covariance matrix $\hat{S}$ quantifies the error of the state vector and is the sum of the independent sources of error: smoothing error $S_s$, representing the retrieval's limited vertical resolution, and the retrieval noise $S_M$, representing the uncertainty due to errors in the measurement. $\hat{S} = S_M + S_s$. The error covariance matrix produced by the retrieval does not include model parameter errors or forward model errors (Frieß et al., 2006). The error covariance matrix is calculated following Eq. (4):

$$\hat{S} = \left(K^T S_\varepsilon^{-1} K + S_a^{-1}\right)^{-1}$$ (4)

$S_E$ and $S_a$ are the measurement and a-priori covariance matrices, respectively. In our retrievals, $S_a$ was determined by setting the relative error of the a-priori to 100%. The $S_E$ was the diagonal matrix of errors of the retrieved dSCDs as determined by the DOASIS retrievals.

**2.3.3 Conversion of Other Instruments' Data for Comparison to MAX-DOAS Data**

Lidar and AERONET extinction data were converted to the MAX-DOAS aerosol retrieval wavelength of 361 nm following Eq. (5):

$$E(361nm) = E(\lambda_1) * \left(\frac{361nm}{\lambda_1}\right)^{-\propto}$$ (5)

Equation (5) accounts for the dependence of aerosol extinction on wavelength based on the Angstrom exponent, $\propto$. AERONET 300-500nm and 340-440nm Angstrom exponents were used to convert the 532 nm lidar aerosol extinctions and the 380 nm and 340 nm AERONET AODs, respectively. The two resulting AERONET AODs at 361 nm were then averaged. The Angstrom exponent was assumed to be constant with altitude and representative of both field sites. The similarity in trends in AODs and trace gas VCDs between the two sites can indicate when the Angstrom exponent determined from Oski-Ôtin was valid for both sites.

Due to the limited vertical resolution of the MAX-DOAS measurements, MAX-DOAS vertical profiles of aerosol extinction and AODs can only be directly compared to lidar profiles and AODs after smoothing the 361 nm lidar profiles using the MAX-DOAS averaging kernel matrix, **A** (Rodgers and Connor, 2003). The lidar AODs referred to in the paper below and shown in plots are the smoothed AODs determined by vertically integrating the smoothed lidar vertical profiles of extinction at 361 nm unless otherwise stated.

The lidar profiles were averaged into the same altitude and temporal intervals as the MAX-DOAS retrievals and then smoothed using the respective matrix **A** following Eq. (6):





$$x_s = x_a + A(x_L - x_a) \qquad (6)$$

Where $x_s$ is the smoothed lidar profile, $x_a$ is the MAX-DOAS retrieval a-priori profile and $x_L$ is averaged lidar profile. $x_s$ represents the (noise-free) vertical profile that the MAX-DOAS retrieval would produce if $x_L$ was the true atmospheric profile given the variable sensitivity of the MAX-DOAS retrieval with altitude. The deviation of $x_s$ from $x_L$ at each altitude depends on $x_a$ and the sensitivity of the MAX-DOAS to the atmosphere at that altitude. MAX-DOAS sensitivity to the true

atmospheric state decreases with increasing altitude (Frieß et al., 2006) with typical height resolutions of ~200 m at lower altitudes, increasing to ~700 m at higher altitudes. Therefore, the smoothing is generally expected to smooth the true profiles towards lower altitudes. Also, even if the two instruments viewed the same air mass, the retrieved and smoothed profiles are expected to differ at least slightly due to two factors. The first factor is the retrieval noise $G\varepsilon$, which is unknown since the true measurement error $\varepsilon$ is unknown. $G$ is the gain matrix, which describes the retrieval's sensitivity to the measurements.

The true smoothed profiles would be described using the following Eq. (7):

$$x_{retrieved} = x_a + A(x_{true} - x_a) + G\varepsilon \qquad (7)$$

The second factor is that lidar vertical profiles observed straight up, measured only above 100 m a.g.l. and are the least sensitive close to the surface. The 0-100 m a.g.l. extinction in the lidar profiles was assumed to be equal to the average extinction measured between 100-200 m a.g.l. but the vertical profiles may have been variable below 150 m a.g.l. Uncertainty in the lidar vertical profiles is greatest in the lowest 150 m a.g.l., introducing uncertainty into the smoothed lidar

profiles.

Error bars on the MAX-DOAS AODs, VCDs and mixing ratios shown in figures were obtained from the optimal estimation retrieval. Error bars on the lidar and AERONET AODs are the standard error of the temporally averaged values since these instruments have a finer time resolution than the MAX-DOAS retrievals. Error bars on the active-DOAS mixing ratios are the root sum square errors of the standard error of the averaged values and the average error reported by the respective

DOASIS retrievals. Error bars on the Pandora VCDs are root sum square errors of the standard error of the average values and the reported instrumental precision. Deming fit linear regressions were performed using the Monte Carlo method, which included the errors on the *x* and *y* data, with the "linfitxy" function in MATLAB (Browaeys, 2017). The Aug 23 and Sep 03 AERONET and Pandora data were also correlated with MAX-DOAS and lidar data by subtracting 30 minutes from the Oski-Ôtin data to account for the time of air mass transport between the Fort McKay South and Oski-Ôtin given the wind-

speeds.

## 3 Results and Discussion

This paper discusses results from largely cloud-free days when industrial plumes were observed ( Aug 23, Sep 03, Sep 04, Sep 06, Sep 07) and one day with clean conditions ( Sep 05). Nine days are not discussed due to the presence of clouds most of the day.





Vertical profiles of wind speed and -direction measured by the windRASS are shown in Figures 2 and 3, respectively. A summary of wind conditions and pollution levels for each day is shown in Table 3.

Aug 23 exhibited the greatest enhancements in aerosol and trace gas pollution during the study. Wind-directions in the morning were North to East-North-East and South-East to South-South-West in the afternoon. Winds were relatively low-
speed with minimal wind shear. The pollution enhancement periods were associated with Southerly (S) winds, suggesting that air-masses rich in industrial emissions originated from the Syncrude and Suncor mining areas south of the sites (Fig. 1). The pollution enhancements impacted both sites (AMS 13 and Oski-Ôtin).

Sep 03 exhibited moderate pollution levels. Pollution data is only presented from 11:00 onwards due to the presence of clouds before this time. Wind-directions varied from South-East to South-South-West with occasional South-West to North-
West winds. Significant wind-shear was observed in the vertical profiles of wind. The pollution enhancements impacted both sites.

Sep 04 exhibited moderate pollution levels. Wind-directions were frequently South to South-Easterly with intermittent periods of South-West and North-West winds. Significant wind shear was observed: wind-directions tended to rotate clockwise from South-South-East near the surface to North-East as altitude increased. The limited afternoon wind data
suggest North-West winds. Wind speeds were low to moderate, tending to increase with altitude. The pollution enhancements impacted both sites.

Sep 05 exhibited the cleanest conditions and greatest wind-speeds during the study. Winds were West-South-West to Westerly. Both sites were impacted by air-masses that passed over boreal forests.

Sep 06 exhibited low to moderate pollution enhancements in the morning with low pollution conditions in the afternoon.
Winds were North to North-Easterly but varied over time and with altitude. Wind-speeds tended to be low at the surface but moderate to large at higher altitudes; significant wind-shear was present. Fort McKay South was impacted by emissions from facilities north of the sites: Shell Jackpine and Muskeg River Mines, CNRL Horizon, and Imperial Oil.

Sep 07 exhibited moderate to low pollution. Winds-directions were South-South-East during the morning and South-West to South-South-West during the afternoon. Significant wind shear was observed in the lowest 400 m a.g.l. between 9:00 and
11:00 and during the afternoon around 16:00. Different air-masses may have impacted the two sites.

### 3.1 Inter-comparisons of MAX-DOAS aerosol retrievals with lidar and AERONET data

The AODs from the MAX-DOAS, lidar and AERONET sun photometer instruments exhibited similar temporal trends on Aug 23, Sep 03, Sep 04, Sep 05 and Sep 07 (Figs. 4-9 (a)). The MAX-DOAS AODs were statistically different from the lidar and AERONET AODs for approximately half the data, even when the two sites experienced the same air-masses. This
result is expected based on three factors: 1) the different vertical extents of the atmosphere observed by the instruments, 2) temporal variability in the lidar S-ratio, and 3) the limited sensitivity of the MAX-DOAS measurements at higher altitudes.



These factors will be discussed below to evaluate the performance of the MAX-DOAS AOD retrievals under various atmospheric conditions.

### 3.1.1 Impact of Instrumental Vertical Sensitivity on AOD

AERONET AODs were generally significantly greater than MAX-DOAS and lidar AODs, except during the greatest pollution events (Figs. 4-9 (a)). During the low-pollution day of Sep 05, AERONET AODs reached a maximum of 0.15±0.00 while maximum MAX-DOAS AODs were 0.08±0.01 (Fig. 4(b)). On Sep 05 the MAX-DOAS and AERONET AODs had a slope of linear correlation of 1.04±0.08 ($R^2$=0.77) but had a linear intercept of -0.08±0.01 km$^{-1}$. This negative intercept can be attributed to aerosol loading above the boundary layer that was observed by the sun photometer but not by

the MAX-DOAS. This result is expected because the AERONET sun photometer observed aerosol extinction throughout the entire column (tropospheric and stratospheric) while the MAX-DOAS and smoothed lidar profiles observed up to 4 km. Further, the MAX-DOAS retrieved and smoothed lidar profiles likely only captured enhancements below 2 km because of the exponentially decreasing a-priori profiles used and the decreasing sensitivity of the MAX-DOAS retrieval with increasing altitude. The MAX-DOAS and smoothed lidar AODs are, therefore, expected to be significantly smaller than the

AERONET AODs when the aerosol extinction in the boundary layer was "clean" and contributed a small fraction to the total tropospheric extinction (e.g., Fig. 10; Aug 23_9:10). MAX-DOAS AODs were also significantly smaller than AERONET AODs even under moderately polluted conditions when the magnitudes of the aerosol extinction remained enhanced above the boundary layer. On Sep 04 the extinctions above the boundary layer could be relatively large, ~1/3 the near-surface extinctions (Figs. 10 & 15 (a)), leading to much smaller MAX-DOAS AODs than AERONET AODs (Fig. 6(a)). Aerosols

can be non-trivial in the free troposphere since fine mode particles can remain in the atmosphere for days (Zhong and Zaveri, 2017). These results indicate that the ratio of the MAX-DOAS AODs to AERONET AODs depends on the location of the aerosol extinction within the tropospheric profile. The use of simple linear regressions to evaluate the performance of MAX-DOAS AOD retrievals using sun photometer AODs may be appropriate only when the aerosol extinction in the boundary layer dominates the total tropospheric AOD.

**3.1.2 Impact of S-ratio Variability on Lidar AODs**

MAX-DOAS AODs significantly exceeded the smoothed lidar AODs during the most polluted periods (Fig. 4(b)). This result is unexpected, given that the instruments' AODs should ideally be equal when the instruments observed the same airmasses. However, the deviation can be explained by variation in the lidar S-ratio. The S-ratio of 25 steradians (sr) appears accurate during relatively clean periods (e.g., Sep 05, Aug 23 morning) but an underestimation under the industrially

polluted conditions of the afternoon of Aug 23 (Figs. 4(b) & 12). Modeled S-ratios for Aug 23 were 21-28 sr during the low pollution morning and 36-44 sr during the peak pollution enhancement at ~16:50 (Table 4). The morning S-ratios were





calculated using the refractive indices of toluene or kaolinite based on the dominance of organic particles and dust in the region during background atmospheric conditions (Fig. 11 (a)). The afternoon S-ratios were modelled using the refractive index of sulphate particles based on the significant enhancement in sulphate particle loading (Fig. 11 (a)). Increased loading of sulphate particles tends to increase the S-ratio. Note that the 16:50 S-ratios were greater than the morning S-ratios for all

refractive indices because the particle size distribution of the industrial plume (fine-mode dominated) increased the S-ratio. The modelled variability in the S-ratio is supported by lidar measurements in the AOSR in 2018 that allowed determination of temporal and vertical variability in the S-ratio (Strawbridge et al., 2018). Measured S-ratios ranged from 20 to 60 sr within the boundary layer at Oski-Ôtin in 2018 (Fig. S6).

Based on these results, lidar vertical profiles of aerosol extinction were retrieved using an S-ratio of 44 sr for the extinction

below the free troposphere after 14:30 on Aug 23. As shown in Figure 4 (a), the updated lidar AODs are in more reasonable agreement with the MAX-DOAS and AERONET AODs compared to the original lidar AODs shown in 4(b). The linear regression of the MAX-DOAS and updated lidar AODs has a slope of 1.15±0.02 with an intercept of -0.01±0.02 ($R^2 = 0.97$) instead of 2.18±0.03 for the original lidar AODs ($R^2 = 0.97$) (Table S3). Modelling S-ratios using particle data measured at the near-surface appears to be valid during Aug 23 because the vertical profile was relatively well mixed. A well-mixed

boundary layer was indicated by the similarity in temporal trends between the Active-DOAS mixing ratios and MAX-DOAS VCDs and between the AODs and the surface particle loading (Figs. 4 (g, h) & 11 (a)). However, if the distribution of particles in space is inhomogeneous, this method cannot be used to determine the S-ratio of the total boundary layer.

Results from Sep 03 and Sep 06 illustrate that near-surface measurements of particles properties can be invalid for modelling the total column S-ratio due to complex vertical profiles of particles. Despite near-surface enhancements in sulphate particles

(Fig. 11) and $SO_2$ mixing ratios observed by Active-DOAS (Fig. 5(f)), the MAX-DOAS and lidar AODs were very similar after 11:30-17:00 on Sep 03 (Fig. 5(a)). The MAX-DOAS and Pandora $SO_2$ VCDs were moderate compared to the enhancements on Aug 23, suggesting that the sulphate enhancements were confined mainly near the surface after 11:30. Due to wind-shear, the near-surface air (<200 m a.g.l.) was often impacted by industrial pollution from the south while the air at higher altitudes was impacted by less polluted regions (North-West and Northerly winds), particularly around 14:00 (Fig. 3

(b)). Thus, the S-ratio of 25 sr was representative of the total boundary layer after 11:30 despite sulphate enhancements at the surface, leading to similar magnitudes of MAX-DOAS and lidar AODs. S-ratios modelled using the near-surface measurements of particles during the afternoon of Sep 03 would have overestimated the S-ratio within the total boundary layer.

Similarly, near-surface measurements of particles would not represent the total boundary layer on Sep 06 due to an elevated

industrial plume. The MAX-DOAS $NO_2$ VCDs remained enhanced while the Active-DOAS mixing ratios rapidly decreased from ~7 ppb to ~1 ppb (Fig. 8(g)). The MAX-DOAS AODs approached the AERONET AODs around noon (Fig. 8(a)), maximizing around the time that the lidar observed elevated vertical profiles of aerosol extinction. These results suggest that elevated plumes from the industrial facilities to the north of Fort McKay South (Figure 1) increased the S-ratios at higher





altitudes. S-ratios modelled using the surface data during this time, therefore, would have underestimated the average S-ratio within the boundary layer.

These results suggest that the MAX-DOAS retrievals of AODs performed well when the vertical extent of instrumental viewing and S-ratio variability are considered.

### 3.1.3 Comparison of MAX-DOAS Vertical Profiles of Aerosol Extinction with Averaged and Smoothed Lidar Vertical Profiles

MAX-DOAS vertical profiles of aerosol extinction are compared to averaged and smoothed lidar vertical profiles in Figures 12 to 18.

Smoothing alters the shape and magnitude of the averaged lidar profiles in several ways. Smoothing the averaged lidar profiles generally "compresses" the profiles by vertically attributing extinction at higher altitudes to lower altitudes (compare panels (a) and (b) in Figs. 12-18). This result is expected due to the decreasing sensitivity of the MAX-DOAS retrieval with increasing altitude apparent in the averaging kernels (Fig. S7). The smoothing also replaces lidar aerosol extinction above ~1.5 km a.g.l. with the (small) a-priori extinction values because the MAX-DOAS measurements have little information content at high altitudes (Fig. S7). This effect is apparent when comparing the Sep 03 averaged and smoothed lidar profiles above 1.5 km a.g.l. (Fig. 14). Profiles that were relatively uniform within a few hundred meters of the surface can sometimes be smoothed into apparently elevated profiles because the averaging kernel attributes much of the extinction from altitudes aloft to one altitude bin closer to the surface. For example, the averaging kernels for the Sep 04 14:10 retrieval for altitudes 0.55 to 1.25 km peak at 0.45 km rather than at their respective height (Figs. S7 & S8). Conversely, the smoothing can transform vertically narrow and distinctly elevated profiles near the surface into exponentially decreasing profiles due to the limited vertical resolution of the retrieval (see the 9:30 profile in Fig. 17). Therefore, interpretation of the retrieved MAX-DOAS profiles must account for the effects of smoothing on the true atmospheric profiles.

On Aug 23, the MAX-DOAS and smoothed lidar vertical profiles (S-ratio = 25 sr) exhibited similar temporal trends and vertical enhancements within the boundary layer (Fig. 12). The magnitudes of aerosol extinctions were consistent between the smoothed lidar and MAX-DOAS vertical profiles in the morning, supporting the hypothesis that the S-ratio of 25 sr was appropriate for "clean" periods. In contrast, the MAX-DOAS extinctions exceeded the smoothed lidar extinctions in the afternoon (Fig. 12). Using an S-ratio of 44 sr within the afternoon plume (discussed in 3.1.2) resulted in smoothed lidar profiles consistent with the MAX-DOAS profiles (Fig. 13). While temporal trends and overall magnitudes were similar, MAX-DOAS retrievals tended to exhibit more distinctly elevated profiles than the smoothed lidar profiles. The use of a constant S-ratio within the plume may have caused the lidar profiles to appear more vertically uniform than the true profiles since S-ratios can maximize where extinction peaks (Fig. S6). Also, the MAX-DOAS viewing geometry observed air masses south of the field site, closer to industrial sources, where the vertical profiles may have been less well-mixed. Finally, MAX-DOAS measurement errors can be mapped into the retrieved profile, leading to uncertainties, but are probably only important at higher altitudes where the measurements contain little information content. Deviations in the MAX-DOAS





profiles from the smoothed lidar profiles after 17:00 can be attributed to reduced light levels and the longer retrieval time, reducing signal-to-noise ratio and the probability of the viewed airmasses changing significantly within the time required to capture the measurements for the retrieval, respectively.

For the Sep 03, Sep 04 and Sep 07 (morning) comparisons, the MAX-DOAS retrieved profiles generally captured the same

temporal and vertical trends in extinction enhancements as the smoothed lidar profiles, but the lidar extinctions were smaller than the MAX-DOAS extinctions (Figs. 14, 15, & 18). The S-ratio of 25 sr probably underestimated the true values given the presence of sulphate particles (Fig. 11 (b, c, f)) and enhanced $SO_2$ VCDs (Figs. 5(b), 6(b) & 9(b)). On Sep 05 the S-ratio of 25 sr is expected to be appropriate due to the clean conditions. The magnitudes of the MAX-DOAS extinctions were unexpectedly greater than the smoothed lidar extinctions but were generally equal within error (Fig. 16). The Sep 06 MAX-

DOAS aerosol retrievals appear noisier than the smoothed lidar profiles (Fig. 17). The elevated plumes present in the MAX-DOAS retrievals but not in the lidar profiles may be related to an increased S-ratio due to the impact of plumes from north (Fig. 8 (g)). On Sep 07 the MAX-DOAS aerosol extinction profiles were of greater magnitude and different in vertical profile shape compared to the smoothed lidar profiles after 12:00. The deviation can be attributed to significant wind-shear and rapid temporal variation in the wind profiles after 12:00 (Figs. 2(f) & 3(f)). Aerosol extinction magnitudes varied by up

to a factor of five within 10 minutes in the afternoon (Fig. S9). These conditions violate two assumptions of the MAX-DOAS retrievals: low horizontal inhomogeneity and that the spectra measured during the retrieval time observed the same airmass. Although the MAX-DOAS retrievals of AOD were consistent with the smoothed lidar AODs, the temporal and vertical resolutions of the MAX-DOAS retrievals were insufficient to retrieve accurate vertical profile shapes. The afternoon MAX-DOAS vertical profile retrievals are, therefore, not expected to represent the true atmospheric state.

**3.2 Evaluation of MAX-DOAS Trace gas Retrievals**

**3.2.1 Comparison of MAX-DOAS and Pandora Trace gas VCDs**

The MAX-DOAS and Pandora $SO_2$ and $NO_2$ VCDs exhibited similar temporal trends over the four days of comparison, except during the afternoon of Sep 07 (Fig. 4(c, d); (b, c) in Figs. 5, 6 & 9).

On Aug 23 the MAX-DOAS and Pandora VCDs were strongly correlated ($R^2 > 0.80$) with linear regression slopes of

$1.55\pm0.07$ and $2.20\pm0.07$ for $SO_2$ and $NO_2$ VCDs, respectively (Table S4). Greater trace gas enhancements were also observed near the surface at Fort McKay South compared to Oski-Ôtin through in-situ measurements by the Wood Buffalo Environmental Association (WBEA) (Wood Buffalo Environmental Association, 2019) (Fig. S10) with slopes of the linear regressions of $1.42\pm0.05$ ($R^2=0.91$) and $1.93\pm0.07$ ($R^2=0.61$) for $SO_2$ and $NO_2$, respectively (Table S2). The strong correlations between the trace gas measurements between the sites indicate that the same airmass impacted both sites but that

a more central (higher concentration) portion of the plume impacted Fort McKay South or significant horizontal dilution of the plume occurred during transport.



The NO$_2$ VCDs had a slope of the regression greater than that of the in-situ NO$_2$ measurements (Table S2). NO$_x$ may have been lost at a faster rate near the surface during transport due to deposition to the surface (e.g., the boreal forests). Transport times between the sites were relatively long (~30 minutes) on this day due to low wind-speeds below 600 m a.g.l. (Fig. 2). NO$_x$ is lost through surface deposition and photochemical conversion to HONO and HNO$_3$ (Finlayson-Pitts et al., 2003;
Wojtal et al., 2011). HONO might be subsequently released as NO and OH, but the HNO$_3$ loss will be virtually permanent.

$$NO_2 + NO_2 + H_2O \rightarrow HNO_{3(aq.surface)} + HONO_{(g)} \tag{8}$$

$$HONO + h\nu \rightarrow NO + OH \tag{9}$$

On Sep 04 the MAX-DOAS and Pandora VCDs exhibited similar temporal trends and were often equal within error (Fig. 6(b, c)). The slopes of the linear correlations of the SO$_2$ and NO$_2$ VCDs were 1.10±0.33 ($R^2$=0.51) and 0.95±0.07 ($R^2$=0.85), respectively (Table S6). The greater variability in SO$_2$ between the sites compared to NO$_2$ is consistent with the in-situ data between Fort McKay South and the WBEA Bertha Ganter site (Fort McKay North; 57.189428, -111.640583) with $R^2$= 0.7 and
0.92 for SO$_2$ and NO$_2$, respectively (Table S6) (Wood Buffalo Environmental Association, 2019). SO$_2$ plumes are more localized in the AOSR, originating mostly from large industrial stacks and fewer sources compared to NO$_2$ (Zhang et al., 2018) (Tables S1 & S2). Note that when MAX-DOAS SO$_2$ VCDs were significantly greater than Pandora SO$_2$ VCDs around noon, the SO$_2$ mixing ratios at Fort McKay South were approximately double those at the Bertha Ganter (Fort McKay) site (Fig. S11). These results suggest that the MAX-DOAS performed well in retrieving accurate VCDs of SO$_2$ despite the
weaker linear correlation with the Pandora VCDs. The two sites appear to have largely experienced the same air-masses within a small temporal period (<20 minutes) due to higher wind-speeds relative to Aug 23 and Sep 03 (Fig. 2 (a, c)). Higher wind speeds likely reduced the maximum enhancements in trace gas VCDs compared to Aug 23 and Sep 03 due to greater dispersion. Wind shear on Sep 04 (Fig. 3 (c)) may also have transported only certain altitudes of the elevated plumes from south of Fort McKay South to the sites. In contrast, wind shear on Aug 23 was limited within 500 m a.g.l. (Fig. 3(a)).
On Sep 03 and Sep 07, the MAX-DOAS and Pandora VCDs demonstrated weak linear correlations ($R^2$ <0.2) (Tables S5 and S9).

The Sep 03 VCD correlations are inconclusive due to the limited number of data points and relatively little variation in the Pandora VCDs. The MAX-DOAS VCDs tended to be higher than the Pandora VCDs. Unlike on Aug 23, an examination of the in-situ data between sites is not helpful due to the significant wind-shear on Sep 03 and the presence of elevated plumes.
Based on the good agreement between the MAX-DOAS and Pandora VCDs on Sep 04 with similar VCD magnitudes, the apparent overestimation could be due to different air-masses experienced by the two sites.

On Sep 07 the MAX-DOAS were similar to the Pandora trace gas VCDs before 13:30 but were much larger after (Fig. 9 (b, c)). The deviation of the MAX-DOAS VCDs is an expected result given the rapid spatial and temporal variation in the wind profiles (discussed in 3.1.3). Errors of the trace gas retrievals can be expected to be even greater than the aerosol retrieval





errors because the retrieved aerosol profiles were used as forward model parameters in the trace gas retrieval. The afternoon MAX-DOAS trace gas retrievals on Sep 07 are not expected to represent the true atmospheric state.

Inter-comparisons of the Pandora and MAX-DOAS VCDs show that the MAX-DOAS retrievals of trace gas VCDs performed well under low to moderate wind-speeds and when vertical profiles of pollution were relatively constant within

the retrieval period.

### 3.2.2 Comparison of MAX-DOAS 0-100 m Retrieval with Active-DOAS Mixing Ratios

The 0-100 m a.g.l. MAX-DOAS trace gas retrievals are shown with the Active-DOAS mixing ratios in Figures 4 (e, f) and Figures 5-9 (d, e). The MAX-DOAS retrievals generally captured the Active-DOAS temporal trends but tended to overestimate the magnitudes. The MAX-DOAS retrieval yields an estimate of the average concentration within the 0-100 m

layer, which is larger than the surface value in case of uplifted layers. Therefore, in-situ near-ground instruments, such as Active-DOAS, are required when accurate surface mixing ratios are required.

The MAX-DOAS retrievals were most consistent with the Active-DOAS measurement during the late afternoon of Aug 23 (Fig. 4 (e, f)). $SO_2$ was at its highest levels and assumed to be relatively well-mixed within the boundary layer based on the similarity in the temporal trends in $SO_2$ VCDs and surface mixing ratios (Fig. 4 (g)) and the uniformity of the lidar vertical

profiles <1 km a.g.l. (Fig. 13 (a)). The mixing ratios were equal within error during the morning and after 14:00 with some differences in the early afternoon that may be due to the different viewing geometry. On days other than Aug 23, the uncertainty in the surface retrieval is often too high for reliable comparison when the near-surface when $SO_2$ and $NO_2$ were <20 ppb and <10 ppb, respectively. Overall, the MAX-DOAS retrievals of 0-1000 m performed well, considering the frequently complex vertical profiles observed during the study.

### 3.2.3 Temporal trends of MAX-DOAS Trace gas VCDs and Active-DOAS Mixing Ratios

Active-DOAS mixing ratios are shown with MAX-DOAS VCDs in Fig. 4(g, h) and in Figs. 5-9 (f, g). The VCDs and mixing ratios exhibited similar temporal trends on Aug 23, Sep 04-06 (Fig. 4(g, h); Figs. 6 & 7 (f, g)), but not on Sep 03 and Sep 07 ((f, g) in Figs. 5 & 9). The similar temporal trends in VCDs and mixing ratios observed on Aug 23 are consistent with the limited vertical wind-shear and low to moderate wind-speeds, as discussed previously. In contrast, the ratio of VCDs to

mixing ratios sometimes varied even during short periods on Sep 04 and Sep 06. If the boundary layer is well-mixed, the Active-DOAS mixing ratios and MAX-DOAS VCDs are expected to have similar temporal trends during short periods since the boundary layer is expected to be effectively constant. On Sep 04, the temporal trends were very similar until ~13:30, when the rapid decrease in trace gas mixing ratios was not reflected in the VCDs (Fig. 6 (f, g)), indicating elevated pollution plumes that are apparent in the lidar measurements (Fig. S7 (a)). These observations are a testament to the ability of MAX-

DOAS to observe elevated pollution plumes not detectable at the surface. The differences in the short-term trends in VCDs and mixing ratios are consistent with the wind profile data around 13:30 on Sep 04, which indicates Westerly to





Northwesterly wind directions <300 m a.gl. that are expected to result in relatively clean air near the surface (Fig. 3(c)). Although measurements of the wind profiles above ~250 m a.g.l. were unavailable, southerly winds aloft are suggested by the trace gas VCDs remaining enhanced until ~15:00. While significant enhancements of trace gas near the surface tend to contribute to enhanced VCDs, the opposite may not always occur: elevated plumes that cause enhanced VCDs may not

result in large surface mixing ratios (Fioletov et al., 2016). The observations in this study indicate that elevated enhancements may also result from vertical wind shear. Techniques for estimating emissions from industrial facilities must account for the possibility that different vertical portions of plumes can be transported in different directions. Such complex pollution conditions require pollution monitoring techniques such as MAX-DOAS that can detect elevated pollution plumes. In addition to being able to observe elevated plumes that are under-sampled by in-situ, ground instruments, MAX-DOAS can

be used to estimate emissions when deployed using the mobile-MAX-DOAS technique (Davis et al., 2019 [*soon to be published for interactive discussion on ACP*]).

### 3.2.4    MAX-DOAS Retrievals of Vertical profiles of $SO_2$ and $NO_2$

MAX-DOAS retrievals of vertical profiles of $SO_2$ and $NO_2$ are shown in Figure 19. Unlike the aerosol profiles, co-located measurements of the trace gas vertical profiles were generally not available. The magnitude and vertical location of the

pollution were highly dependent on wind direction and wind shear. The greatest trace gas enhancements occurred under South-South-Easterly wind-directions (Figs. 3 & 19) where pollution originated from the greatest sources of $SO_2$ and $NO_2$ to the South (Fig. 1; Tables S1 & S2). The MAX-DOAS retrievals performed well in terms of the profile shapes expected based on the wind profiles or evidence of elevated plumes. For example, trace gas pollutants in the MAX-DOAS retrievals were confined largely to <200 m on the mornings of Sep 04 and Sep 07 (Fig. 19 (c) & (f)) as expected from the wind-shear (Fig.

3). The elevated profiles of $SO_2$ on Sep 03 before noon and during the afternoon of Sep 04 are consistent with the results discussed previously.

Aircraft measurements of trace gases on Sep 03 allow some comparison of the MAX-DOAS retrieved profiles. A vertical profile of $SO_2$ measured during an aircraft spiral ascent at ~14:27 in the vicinity of Fort McKay South (Fig. 20) is consistent in magnitude and shape with the MAX-DOAS retrieved vertical profile for 11:00-11:20 (Fig. 20). The MAX-DOAS 11:10

profile was used for comparison because it appears to have observed the same plume as the aircraft spiral. Although these two profiles cannot be directly compared due to the differences in time and vertical resolutions, the aircraft profile indicates that the magnitudes and elevated shape of the MAX-DOAS profiles of $SO_2$ are reasonable. The elevated $SO_2$ plumes measured by the aircraft and MAX-DOAS could have originated from upgrader stacks at either the Syncrude or Suncor facilities south of Fort McKay South. The aircraft also passed over Fort McKay South at 16:32, measuring 30 ppb of $SO_2$

and 5 ppb of $NO_2$ at 395 m a.g.l. The MAX-DOAS retrieval for 16:20-16:40 had maximum $SO_2$ values of 57 (±19) ppb at 350 m and maximum $NO_2$ values of 10 (±5) ppb at 650 m. Note that the Active-DOAS measured 20 (±0.1) ppb of $SO_2$ and



4.3 (±0.1) ppb of $NO_2$ near the surface. These measurements, therefore, suggest that elevated plumes were present and that the MAX-DOAS retrieved magnitudes are reasonable.

### 3.3 Advantages of MAX-DOAS

MAX-DOAS has an advantage over the zenith lidar technique in detecting aerosol extinction since lidar retrievals cannot
detect close to the surface due to challenges with signal overlap (Zieger et al., 2011). Quantifying aerosol extinction from lidar measurements also requires additional knowledge (i.e., the S-ratio) (Wagner et al., 2004), as has been highlighted in this paper. The advantage of the MAX-DOAS over the sun photometer (in direct-sun viewing mode) is the ability to determine vertical profiles of pollutants versus only total columns. The MAX-DOAS is complementary to Active-DOAS and other point-source measurements when pollution within the boundary layer is vertically inhomogeneous (see 3.2.3). While
surface level, local measurements of pollutants are often important for applications such as health exposure studies, they may fail to provide the full picture of the total boundary layer pollution. Such in-situ measurements provide highly localized information with little information about elevated plumes that may mix down to the surface down-wind. MAX-DOAS allows remote sensing of airmasses over longer path-lengths, even if plumes are elevated. The MAX-DOAS method is advantageous over satellite measurements when plumes are localized and can provide more information on near-surface
trends.

### 3.4 Limitations of the Inter-Comparisons in this Study

A limitation to validating the MAX-DOAS AODs against lidar and sun photometer data was the different viewing geometry and slightly different locations. Also, Angstrom exponents used to convert the lidar extinctions to the MAX-DOAS retrieval wavelength would ideally be measured at Fort McKay South. Application of a single S-ratio modelled from particle
measurements from the near-surface to the entire lidar vertical profile can introduce errors since the S-ratio may vary vertically (see 3.1.2 and Fig. S6). The S-ratio can be significantly non-uniform with altitude when the vertical profile is composed of layers of anthropogenic (urban, biomass burning), and/or biogenic aerosols or mixtures of them. Even if a layer is well-mixed, the lidar ratio can change with height if the vertical profile of relative humidity is non-uniform (Weitkamp, 2005).
The MAX-DOAS trace gas VCDs should ideally be compared with a co-located Pandora instrument given the possibility of horizontal inhomogeneity between the sites. Validation of the MAX-DOAS 0-100 m retrieval using the Active-DOAS mixing ratios was complicated by the lowest viewing elevation angle observing 5 m above the Active-DOAS light path. The MAX-DOAS "surface" retrieved values are only expected to be equal to the Active-DOAS values when the air masses were well-mixed within 0-100 m a.g.l. A more thorough validation of the MAX-DOAS near-surface retrievals could be achieved
with trace gas measurements at multiple heights within 100 m a.g.l. from a tall tower.



## 4 Summary

In this study, data from a diverse range of instruments have allowed an expansive characterization of the MAX-DOAS retrievals of aerosol extinction, $NO_2$ and $SO_2$. The retrievals performed well at capturing the aerosol loading within the boundary layer. The exception was under conditions of rapid variation in the vertical profiles of pollutants during the retrieval period. The ratio of the MAX-DOAS to sun photometer AODs depended on the vertical location of the aerosol extinction within the atmospheric column. Direct inter-comparisons of AODs between instruments must account for the relative spatial extents observed. The comparison of MAX-DOAS and lidar data combined with S-ratio modelling indicated that accurate S-ratio values are essential to retrieve accurate profiles of aerosol extinction from lidar measurements when particle composition or size distribution varies significantly temporally or spatially. Direct comparison of MAX-DOAS and lidar AODs should be made with caution when knowledge of the S-ratio value(s) is limited. S-ratios can be estimated from measurements of particle size distribution and composition using Mie scattering modelling. However, near-surface measurements of particles should only be used to model S-ratios when the boundary layer is well mixed. Lidar extinction profiles should ideally be determined using a technique that accounts for the vertical and temporal variation in the S-ratio such as in Strawbridge et al. (2018). When the S-ratio variability was accounted for, the results of this study indicate that the MAX-DOAS retrievals of aerosol extinction performed well compared to the smoothed lidar results.

Comparisons of averaged and smoothed lidar profiles of aerosol extinction indicated that the vertical sensitivity of the MAX-DOAS retrievals smoothed the true atmospheric profiles towards the surface. This smoothing can transform vertical profiles that are relatively uniform within the boundary layer into apparently elevated profiles and vice versa. This shape change depends on the location of extinction within the true vertical profile and the averaging kernel matrix of the retrieval. Interpretation of the shape of the MAX-DOAS vertical profiles must account for the instrument's sensitivity to the true vertical profile (i.e., the averaging kernel matrix).

MAX-DOAS retrievals of $NO_2$ and $SO_2$ VCDs performed well in comparison to the Pandora VCDs. The exception was when the aerosol retrievals were inaccurate due to rapidly varying vertical profiles. This was an expected result since the aerosol retrievals are used as forward model parameters in the trace gas retrieval. The MAX-DOAS trace gas retrievals within 0-100 m a.g.l. captured the temporal trends observed by the Active-DOAS measurements, but the MAX-DOAS mixing ratios were statistically greater than the Active-DOAS values, particularly when $SO_2$ and $NO_2$ were <20 ppb and <10 ppb, respectively. Differences between the instruments' values can be attributed to variability in the trace gas profiles within 150 m a.g.l. The MAX-DOAS observed elevated enhancements of pollution undetected by ground-based techniques such as the Active-DOAS, perhaps its greatest asset. Pollution enhancements at surface-level did not always coincide with total boundary layer enhancements, and vice versa, due to elevated plumes and/or significant wind-shear. The MAX-DOAS vertical profiles of trace gases were consistent with the profiles expected based on the wind direction and -shear conditions. Aircraft measurements of $SO_2$ near Fort McKay South on Sep 03 indicated that the magnitudes and elevated shape of the retrievals were reasonable.



A major advantage of the MAX-DOAS technique is the ability to simultaneously retrieve total column and vertical profiles of trace gases and aerosol extinction from spectral measurements without requiring prior knowledge of the aerosols or the vertical profiles of trace gases. These advantages are important in industrial regions where the vertical profiles of pollutants vary temporally and spatially, and in-situ monitoring can under-sample plumes. In the AOSR and similar industrial regions, 5 a full understanding of the air quality conditions requires instruments, such as MAX-DOAS, capable of observing the total boundary layer on a horizontal scale of a few kilometers, in addition to traditional in-situ instruments.

*Author Contributions.* ZYWD: MAX-DOAS study concept, design, investigation and data analysis, data visualization, and writing of manuscript and modifications of the same with contribution from all co-authors. UF: supervision and validation of 10 MAX-DOAS data analysis. KBS: ground-based lidar study concept, design, investigation, and data analysis. MA: airborne lidar study concept, design, investigation, and analysis and S-ratio modelling. SB: analysis and visualization of $SO_2$ flight data. EGS: SMPS design, investigation, and data analysis. AL: active-DOAS investigation and data analysis. VEF: Pandora study concept, design, investigation, and data analysis. IA: AERONET AOD study concept, design, investigation, and data analysis. CAM: PRATMO modelling for Pandora data analysis and provision of Pandora data. JW: supervision of airborne 15 lidar study. MDW and AKYL: SP-AMS study concept, design, and investigation. JB: Project administration and supervision for all studies in the field. JO: SMPS supervision and APS data analysis. JO'B: $SO_2$ flight data study concept, design, and investigation. RS: windRASS study concept, design, investigation, and data analysis. HDO: APS supervision. CM: APS investigation. RM: Active-DOAS study concept, design, and supervision and MAX-DOAS supervision.

20 *Acknowledgements.* Funding for this study was provided by Environment and Climate Change Canada and the Canada-Alberta Oil Sands Monitoring program. Zoe Davis, Akshay Lobo and Sabour Baray acknowledge the financial support provided by the Natural Sciences and Engineering Research Council of Canada (NSERC) Collaborative Research and Training Experience Program (CREATE) Integrating Atmospheric Chemistry and Physics from Earth to Space (IACPES).





**Appendix A** List of Acronyms used in this paper.

| Acronym | Expansion |
| --- | --- |
| A.g.l. | Above ground level |
| AERONET | Aerosol Robotic Network |
| AOD | Aerosol Optical Depth |
| AOSR | Athabasca Oil Sands Region |
| APS | Aerodynamic Particle Sizer |
| BrO | Bromine Oxide |
| CCD | Charge-Coupled Device |
| ClO | Chlorine Oxide |
| DOAS | Differential Optical Absorption Spectroscopy |
| dSCD | Differential Slant Column Density |
| ECCC | Environment and Climate Change Canada |
| FRS | Fraunhofer Reference Spectrum |
| HCHO | Formaldehyde |
| Lidar | Light Detection and Ranging |
| MAX-DOAS | Multi-Axis Differential Optical Absorption Spectroscopy |
| $NH_3$ | Ammonia |
| $NO_2$ | Nitrogen Dioxide |
| $NO_3$ | Nitrate Radical |
| $NO_x$ | Nitrogen Oxides ($NO_2 + NO$) |
| $O_4$ | Tetraoxygen |
| OH | Hydroxyl Radical |
| $PM_{2.5}$ | Particulate Matter with a Diameter < 2.5 Micrometres |
| ppb | Parts Per Billion |
| ppt | Parts Per Trillion |
| RI | Refractive Index |
| SMPS | Scanning Mobility Particle Sizer |
| $SO_2$ | Sulphur Dioxide |



| SP-AMS | Soot-Particle Aerosol Mass Spectrometer |
| UV | Ultraviolet |
| VCD | Vertical Column Density |
| WBEA | Wood Buffalo Environmental Association |

**Table 1** Description and locations of the study instruments.

| Instrument | Variables Measured | Institution | Temporal Resolution | Viewing Direction | Field Site | Reference |
|---|---|---|---|---|---|---|
| Mini-MAX-DOAS | Vertical profiles of $SO_2$, $NO_2$, aerosol extinction | York University | 20-30 minutes | SSE at multiple elevation angles | Fort McKay South | Current Paper |
| Active-DOAS | Mixing ratios of $SO_2$, $NO_2$ | York University | ~2 minutes | SE, horizontal | Fort McKay South | (McLaren et al., 2010) |
| Pandora Sun Photometer | VCDs of $SO_2$, $NO_2$ | ECCC | ~1 minute | Direct sun viewing | Oski-Ôtin | (Fioletov et al., 2016) |
| Sun Photometer | AOD, Angstrom Exponent | ECCC | ~3 minutes | Direct sun viewing | Oski-Ôtin | (Sioris et al., 2017) |
| Ground-based lidar | Vertical Profile of Aerosol Extinction | ECCC | 1 minute | Zenith Viewing | Fort McKay South | (Strawbridge, 2013) |
| TSI APS 3321 | $PM_{10-1}$ Size Distribution | University of Calgary | 6 minutes | N/A | Fort McKay South | (Tokarek et al., 2018) |
| TSI SMPS (3081 DMA, 3776 CPC) | $PM_1$ size distribution | University of Alberta | 6 minutes | N/A | Fort McKay South | (Tokarek et al., 2018) |
| Aerodyne SP-AMS | rBC, $NH4^+_{(p)}$, $SO_4^{2-}{}_{(p)}$, $NO_3^-{}_{(p)}$, $Cl^-{}_{(p)}$, organics | University of Toronto and ECCC | ~1 minute | N/A | Fort McKay South | (Lee et al., 2019) |





| Scintec model MFAS windRASS | Vertical profile of wind and temperature | ECCC | 15 minutes | Zenith Viewing | Oski-Ôtin | (Gordon et al., 2017) |
| Airborne Thermo Scientific 43iTLE | Mixing ratios of $SO_2$ | ECCC | 1 second | N/A | N/A | (Baray et al., 2018) |

**Table 2** Information on MAX-DOAS spectral fitting.

| Gas | Fitting Window | Included in the Fit |
|---|---|---|
| $NO_2$ | 410-435nm | FRS, Ring, Bogumil 2003 $NO_2$ (293K) and Bogumil 2003 (293K and 243K) $O_3$, $3^{rd}$ order polynomial |
| $SO_2$ | 310.5-324nm | FRS, Ring, Bogumil 2003 $SO_2$ (293K) and Bogumil 2003 (293K and 223K) $O_3$, $3^{rd}$ order polynomial, Offset Function |
| $O_4$ | 350-375nm | FRS, Ring, Hermans 2011 $O_4$ Bogumil 2003 (293K) $NO_2$, Bogumil 2003 (293K and 223K) $O_3$, $3^{rd}$ order polynomial |





**Table 3** Daytime wind and pollution conditions during the study days.

| Date | Wind-Directions | Wind-Speeds | Wind-Shear | Pollution Levels |
|------|-----------------|-------------|------------|------------------|
| Aug 23 | Morning: N to ENE<br>Afternoon: SE to SSW | Low | Minimal | Low to very high |
| Sep 03 | Variable; mostly SE to SSW | Low to Moderate | Significant | Moderate to High |
| Sep 04 | Mostly S to SE | Low to Moderate | Significant | Low to moderate |
| Sep 05 | WSW to W | High | Moderate | Very Low |
| Sep 06 | N to NE | Low near the surface, high aloft | Significant | Low to moderate |
| Sep 07 | Morning: SSE<br>Afternoon: SW to SSW | Morning: low<br>Afternoon: moderate to high | Significant | Low to moderate |



**Table 4** Modelled lidar S-ratios (sr) for selected periods on Aug 23 using refractive indices (RI) of different particles.

| Local Time | RI of Toluene | RI of Kaolinite | RI of Sulphate Aerosol |
|---|---|---|---|
| 9:10 | 21 | 25 | 30 |
| 9:30 | 25 | 28 | 34 |
| 14:10 | 17 | 33 | 38 |
| 14:30 | 18 | 33 | 37 |
| 16:30 | 31 | 32 | 38 |
| 16:50 | 36 | 40 | 44 |
| 17:15 | 36 | 40 | 44 |



**Figure 1** Location of field sites Fort McKay South and Oski-Ôtin and major industry sources.





**Figure 2** Vertical profiles of wind speed: Aug 23 (A), Sep 03 (B), Sep 04 (C), Sep 05 (D), Sep 06 (E), and Sep 07 (F).

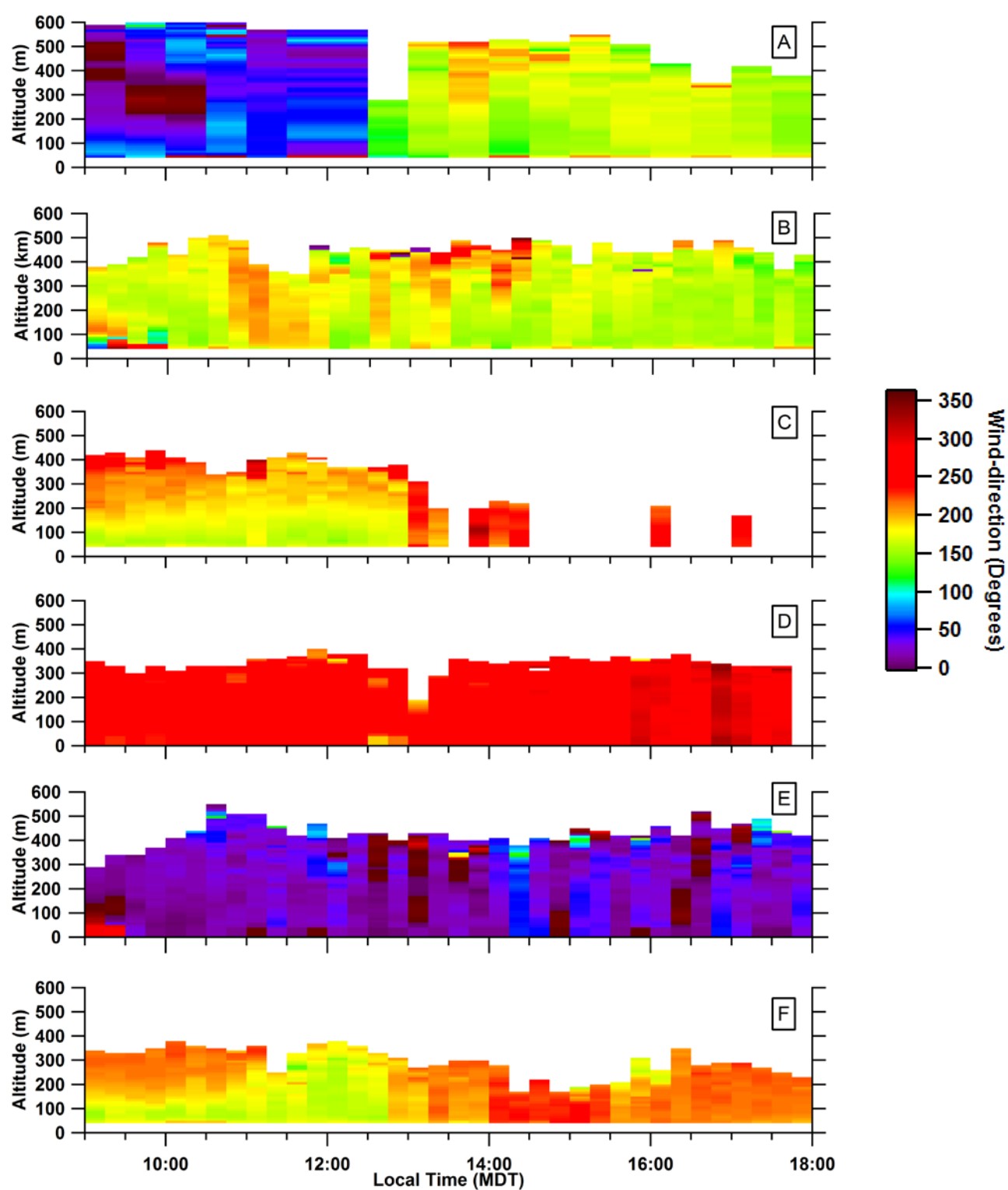

**Figure 3** Vertical profiles of wind direction: Aug 23 (A), Sep 03 (B), Sep 04 (C), Sep 05 (D), Sep 06 (E), and Sep 07 (F).





**Figure 4** Aug 23 AODs from MAX-DOAS, lidar (S-ratio=44 sr >14:30), and AERONET (-30 mins.) (a); AODS from MAX-DOAS, lidar (S-ratio = 25 sr), and AERONET (b); MAX-DOAS and Pandora SO$_2$ VCDs (c); MAX-DOAS and Pandora NO$_2$ VCDs (d); MAX-DOAS 0-100 m and Active-DOAS SO$_2$ mixing ratios (e); MAX-DOAS 0-100 m and Active-DOAS NO$_2$ mixing ratios (f); MAX-DOAS VCDs and Active-DOAS mixing ratios of SO$_2$ (g); and MAX-DOAS VCDs and Active-DOAS mixing ratios of NO$_2$ (h).





**Figure 5** Sep 03 AODs from MAX-DOAS, lidar, and AERONET (a); MAX-DOAS and Pandora SO$_2$ VCDs (b); MAX-DOAS and Pandora NO$_2$ VCDs (c); MAX-DOAS 0-100 m and Active-DOAS SO$_2$ mixing ratios (d); MAX-DOAS 0-100 m and Active-DOAS NO$_2$ mixing ratios (e); MAX-DOAS VCDs and Active-DOAS mixing ratios of SO$_2$ (f); and MAX-DOAS VCDs and Active-DOAS mixing ratios of NO$_2$ (g).

**Figure 6** Sep 04 AODs from MAX-DOAS, lidar, and AERONET (a); MAX-DOAS and Pandora $SO_2$ VCDs (b); MAX-DOAS and Pandora $NO_2$ VCDs (c); MAX-DOAS 0-100 m and Active-DOAS $SO_2$ mixing ratios (d); MAX-DOAS 0-100 m and Active-DOAS $NO_2$ mixing ratios (e); MAX-DOAS VCDs and Active-DOAS mixing ratios of $SO_2$ (f); and MAX-DOAS VCDs and Active-DOAS mixing ratios of $NO_2$ (g).





**Figure 7** Sep 05 AODs from MAX-DOAS, lidar, and AERONET (a); MAX-DOAS SO$_2$ VCDs (b); MAX-DOAS NO$_2$ VCDs (c); MAX-DOAS 0-100 m and Active-DOAS SO$_2$ mixing ratios (d); MAX-DOAS 0-100 m and Active-DOAS NO$_2$ mixing ratios (e); MAX-DOAS VCDs and Active-DOAS mixing ratios of SO$_2$ (f); and MAX-DOAS VCDs and Active-DOAS mixing ratios of NO$_2$ (g).



**Figure 8** Sep 06 AODs from MAX-DOAS, lidar, and AERONET (a); MAX-DOAS SO₂ VCDs (b); MAX-DOAS NO₂ VCDs (c); MAX-DOAS 0-100 m and Active-DOAS SO₂ mixing ratios (d); MAX-DOAS 0-100 m and Active-DOAS NO₂ mixing ratios (e); MAX-DOAS VCDs and Active-DOAS mixing ratios of SO₂ (f); and MAX-DOAS VCDs and Active-DOAS mixing ratios of NO₂ (g).





**Figure 9** Sep 07 AODs from MAX-DOAS, lidar, and AERONET (a); MAX-DOAS and Pandora $SO_2$ VCDs (b); MAX-DOAS and Pandora $NO_2$ VCDs (c); MAX-DOAS 0-100 m and Active-DOAS $SO_2$ mixing ratios (d); MAX-DOAS 0-100 m and Active-DOAS $NO_2$ mixing ratios (e); MAX-DOAS VCDs and Active-DOAS mixing ratios of $SO_2$ (f); and MAX-DOAS VCDs and Active-DOAS mixing ratios of $NO_2$ (g).





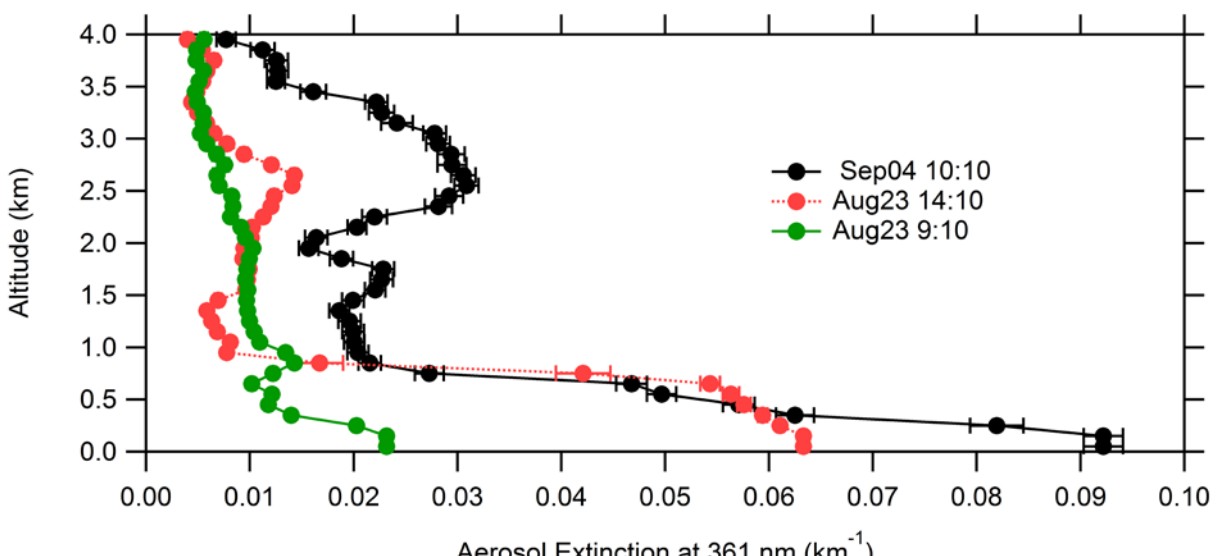

**Figure 10** Examples lidar vertical profiles of aerosol extinction (averaged into MAX-DOAS retrieval height intervals and times) on Aug 23 and Sep 04.





**Figure 11** Near-surface particle compositions on Aug 23 (A), Sep 03 (B), Sep 04 (C), Sep 05 (D), Sep 06 (E), and Sep 07 (F). Note different y-axis scale for Aug 23 and that Nitrate and Refractory Black Carbon are shown multiplied by 10.

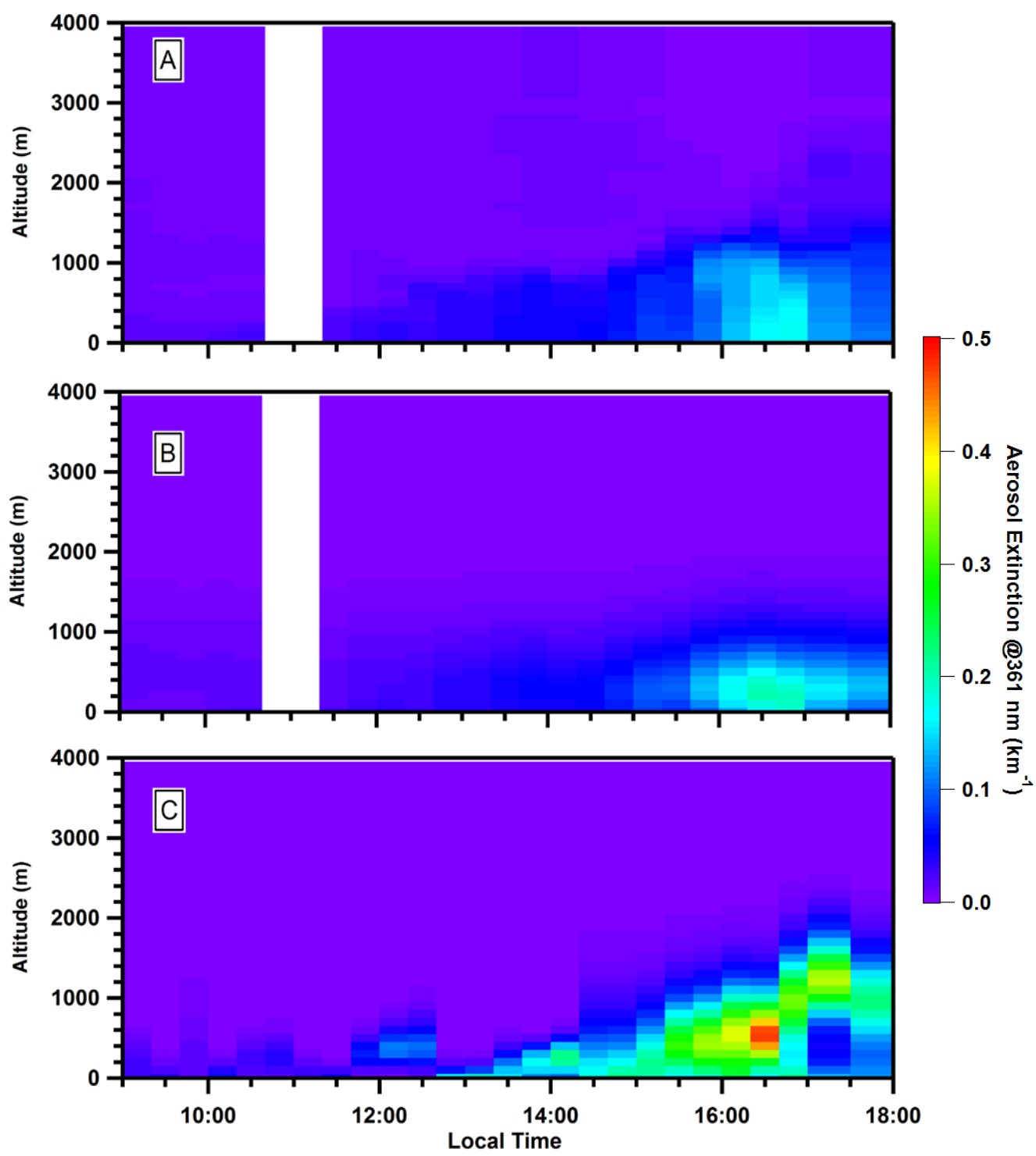

**Figure 12** Aug 23 vertical profiles of aerosol extinction (361 nm) from S-ratio=25 sr: averaged lidar (a), smoothed lidar (b), and MAX-DOAS retrieved (c).







**Figure 13** Aug 23 vertical profiles of aerosol extinction (361 nm) from S-ratio=44 sr within the plume >14:30 local time: averaged lidar (a), smoothed lidar (b), and MAX-DOAS retrieved (c).

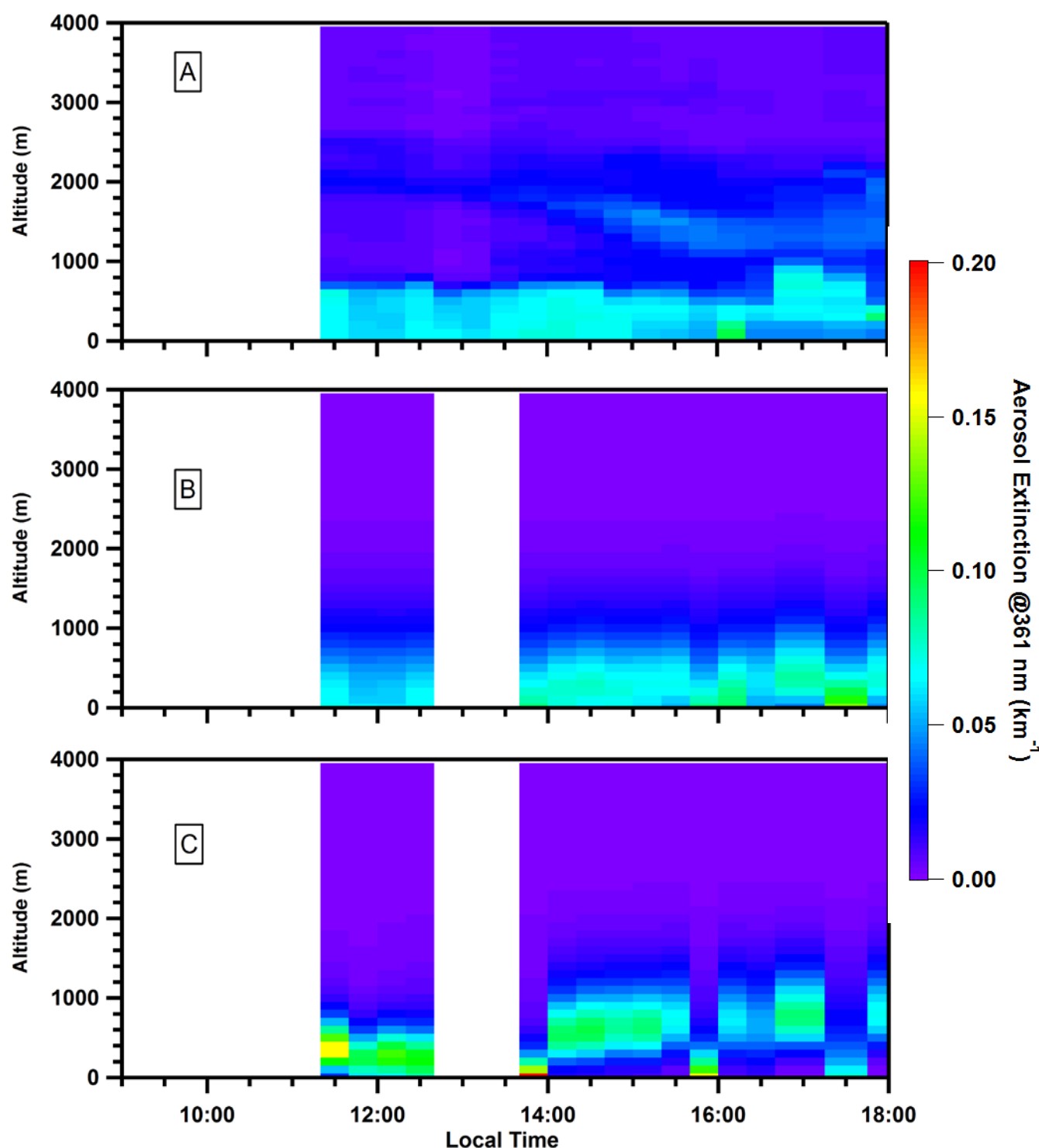

**Figure 14** Sep 03 vertical profiles of aerosol extinction (361 nm) from averaged lidar (a), smoothed lidar (b), and MAX-DOAS retrieved (c).

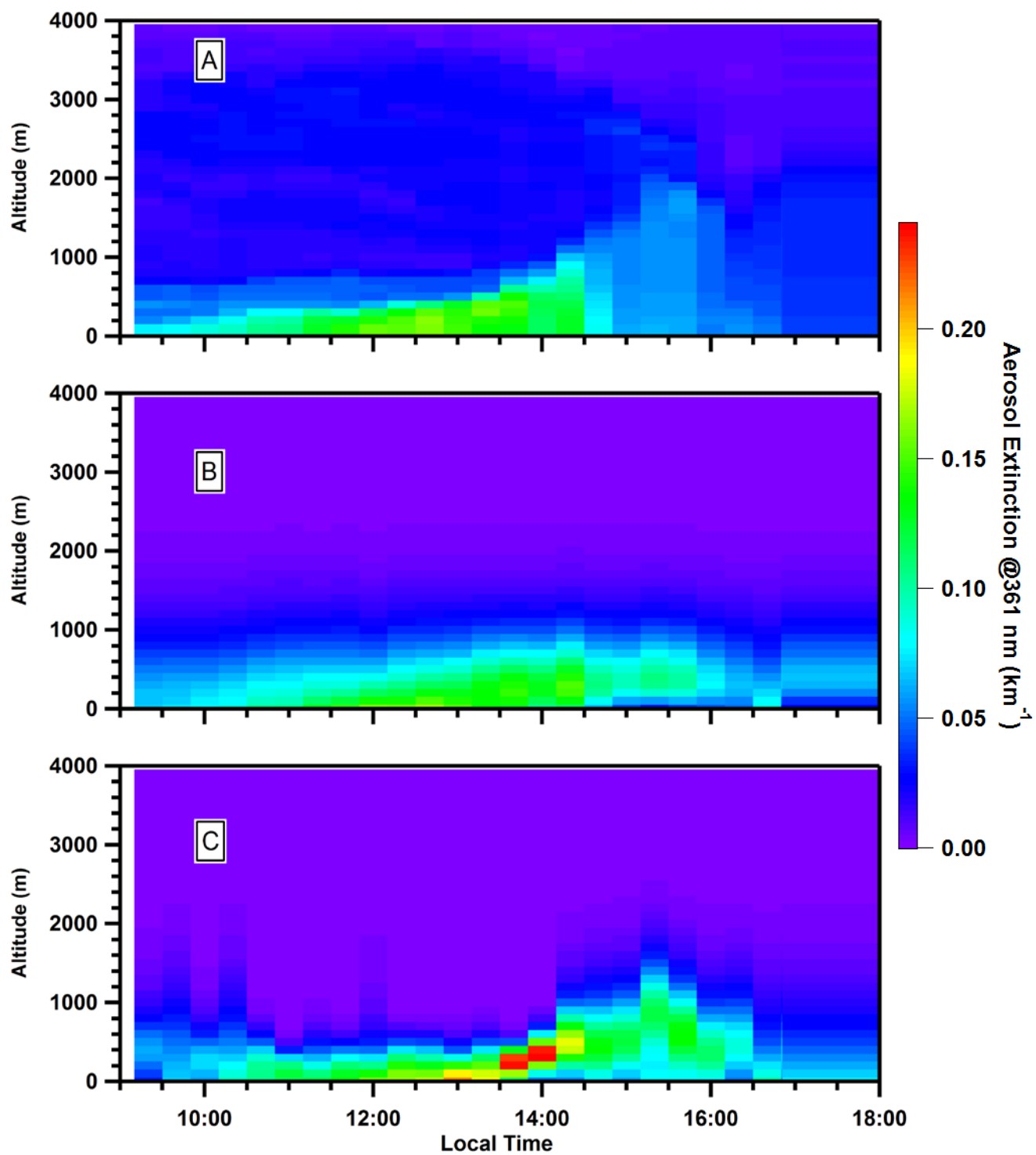

**Figure 15** Sep 04 vertical profiles of aerosol extinction (361 nm) from averaged lidar (a), smoothed lidar (b), and MAX-DOAS retrieved (c).



**Figure 16** Sep 05 vertical profiles of aerosol extinction (361 nm) from averaged lidar (a), smoothed lidar (b), and MAX-DOAS retrieved (c). Omitted data in the afternoon were measurements of cirrus clouds.



**Figure 17** Sep 06 vertical profiles of aerosol extinction (361 nm) from averaged lidar (a), smoothed lidar (b), and MAX-DOAS retrieved (c).





**Figure 18** Sep 07 vertical profiles of aerosol extinction (361 nm) from averaged lidar (a), smoothed lidar (b), and MAX-DOAS retrieved (c).





**Figure 19** MAX-DOAS vertical profiles of $SO_2$ (left column) and $NO_2$ (right column): Aug 23 (A), Sep 03 (B), Sep 04 (C), Sep 05 (D), Sep 06 (E), and Sep 07 (F). Note the different colour scale maximum for Aug 23 and Sep 03.





**Figure 20** Sep 03 vertical profiles of SO$_2$ (ppb) from an aircraft spiral measurement (14:26-14:28 local time) and MAX-DOAS retrieved SO$_2$ vertical profile (local time 11:10). Aircraft spiral shown in Google Earth plot (bottom).



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
