# Peer review of "Validation of MAX-DOAS retrievals of aerosol extinction, SO2 and NO2 through comparison with lidar, sun photometer, Active-DOAS and aircraft measurements in the Athabasca Oil Sands Region."

_Atmospheric Measurement Techniques, 2019_

## Referee Comment (RC1) · Anonymous Referee #1 · 14 Nov 2019

The multi-instrument study demonstrates the value of the MAX-DOAS technique for aerosol and trace gas retrievals as compared to LIDAR, aircraft-based, in situ, and other ground-based measurements. This text is a good example of a well thought out and executed intercomparison (including noted limitations to the comparisons). This is a thoroughly documented study, likely to form part of a thesis (if it hasn't already).

---

## Referee Comment (RC2) · Anonymous Referee #2 · 18 Nov 2019

This paper presents vertical profiles of aerosols, NO2, and SO2 retrieved from MAX-DOAS measurements at the Fort McKay South field site (Alberta, Canada). This site is located close two mining plants which are major sources of industrial pollution in this region. The MAX-DOAS retrieval results are compared to co-located ancillary observations from lidar, AERONET, active DOAS, Pandora, and airborne in-situ analysers instruments. These comparisons based on data sets from various techniques provide a unique opportunity to investigate the performance of MAX-DOAS retrievals under

varying atmospheric conditions in an industrial area.

The manuscript is well written and clearly structured, and presents very interesting results which fit well with the scope of ACP. I recommend the final publication of the manuscript after addressing the following specific comments and technical corrections:

Specific comments:

1/Page 6, lines 1-12: It is not clear in which direction the active-DOAS measurements are performed. Is it the same direction as the MAX-DOAS instrument ? Maybe this information could be added in Figure 1.

2/End of page 7-beginning of page 8: You should add Wagner et al., Atmos. Meas. Tech. (2019) in the list of references on the O4 scaling factor. No O4 scaling is used in the present study. Did you perform sensitivity tests on your aerosol retrievals and you came to the conclusion that a scaling factor was not needed ? Or you simply decided not to use any scaling factor ? I think this should be further discussed in the paper and sensitivity test results could be also added to make the study more robust (e.g. what is the impact of a scaling factor on the agreement with AERONET data ?).

3/Page 9, line 3: According to Rodgers (2000), weighting function K should be equal to $\delta y/\delta x$ and not $\delta F/\delta x$.

4/Page 9: I think you should justify your choice of aerosol extinction and trace gas concentration a priori profiles. Did you perform sensitivity tests for the selection of these a priori profiles, especially in terms of scaling height ?

5/Page 9, lines 30-31: To my knowledge the SCIATRAN RTM is not based on a Monte Carlo approach. This point should be clarified.

6/Page 10, line 10: The relative error of the a priori was set to 100% for the construction of the Sa matrix. Did you set the extra-diagonal terms to zero and, again did you perform sensitivity tests for the selection of this relative error value. Also related: nothing is said in the paper about the quality control of your MAX-DOAS retrievals. For instance,

what are the typical degrees of freedom for signal (DOFS) values of your aerosol and trace gas MAX-DOAS retrievals and what is the level of agreement between measured DSCDs and those modelled using the retrieved profiles as input ?

7/ Comparisons between MAX-DOAS and AERONET AODs (Figs 4a-9a): MAX-DOAS significantly underestimates (by sometimes more than a factor of 2) AERONET AODs. What happens if you used a different a priori profile with a scaling height larger 0.6 km (e.g. 1.2 km) ? Does it improve the agreement with AERONET without degrading the quality of your retrieval (see point 6/) ? Also, is the application of a O4 scaling factor can improve the agreement with AERONET ?

8/Page 22, lines 1-3: You said that a major advantage of the MAX-DOAS technique is the simultaneous retrievals of total columns and vertical profiles of trace gases and aerosol extinctions without requiring a priori information. I am a bit puzzled by this sentence since a priori information is needed in the Optimal Estimation retrieval approach you used. A clarification is needed here.

Technical corrections:

'->' denotes 'should be replaced by'

1/'Honninger et al.' -> 'Hönninger et al.'

2/'Clemer et al.' -> 'Clémer et al.'

3/The first sentences of both Sections 2.1 and 2.2 are a bit redundant. I would start Section 2.1 by 'The MAX-DOAS instrument was operated at an elevation of $\sim$10 m. ...' and Section 2.2 by 'The MAX-DOAS instrument is a mini-DOAS spectrometer from Hoffmann Messtechnik GmbH measuring scattered sunlight. ...' or something similar.

4/Page 5, line 25: 5°C (C not in superscript)

5/Page 22, line 1: 'total column' -> 'total columns'

6/List of References: 'Atmospheric Meas. Tech.' -> 'Atmos. Meas. Tech.'; Same for

'Atmospheric Chem. Phys.'

7/Legends of Figures 4-9: You should add a short description of the error bars presented in the plots.

---

## Author Comment (AC1) · 30 Dec 2019

The authors thank referee #1 for taking the time to read and comment on the discussion paper. We presume that no changes are called for at this time.

---

## Author Response (AR1)

**Response to Editor.**

Our responses to referee comments are highlighted in blue below. Referee #1 seemed to be supportive and did not call for any changes.

**Response to Interactive Comment from Referee #2**

**General Comment**

This paper presents vertical profiles of aerosols, NO2, and SO2 retrieved from MAXDOAS measurements at the Fort McKay South field site (Alberta, Canada). This site is located close two mining plants which are major sources of industrial pollution in this region. The MAX-DOAS retrieval results are compared to co-located ancillary observations from lidar, AERONET, active DOAS, Pandora, and airborne in-situ analysers instruments. These comparisons based on data sets from various techniques provide a unique opportunity to investigate the performance of MAX-DOAS retrievals under varying atmospheric conditions in an industrial area.

The manuscript is well written and clearly structured, and presents very interesting results which fit well with the scope of ACP. I recommend the final publication of the manuscript after addressing the following specific comments and technical corrections:

Response: The authors thank referee #2 for reading and commenting on the discussion paper. We are pleased that it meets with your approval, pending the specific comments and technical corrections below.

**Specific Comments**

**1.** Page 6, lines 1-12: It is not clear in which direction the active-DOAS measurements are performed. Is it the same direction as the MAX-DOAS instrument ? Maybe this information could be added in Figure 1.

**Response: Direction of active-DOAS light path added to Page 6 lines 2-3.**

**2.** End of page 7-beginning of page 8: You should add Wagner et al., Atmos. Meas. Tech. (2019) in the list of references on the O4 scaling factor. No O4 scaling is used in the present study. Did you perform sensitivity tests on your aerosol retrievals and you came to the conclusion that a scaling factor was not needed ? Or you simply decided not to use any scaling factor ? I think this should be further discussed in the paper and sensitivity test results could be also added to make the study more robust (e.g. what is the impact of a scaling factor on the agreement with AERONET data ?).

**Response**: The Wagner et al. 2019 reference was added on page 8 line 5. A statement that the scaling factor was not used because of a lack of conclusive need based on the literature and the

good agreement between the lidar and MAX-DOAS AODs when the modelled S-ratios were applied on Aug 23 was added on page 8 lines 7-9.

Note that while the lidar and MAX-DOAS AODs are expected to be equal within error because the 0-4 km lidar profiles of aerosol extinction were smoothed with the MAX-DOAS retrieval information, the AERONET AODs are expected to be greater than both because the sun photometer observed the entire atmospheric column. Since the MAX-DOAS AODs retrievals used an exponentially decreasing a-priori that reduced to near-zero (<0.005) extinction above 2 km (chosen for reasons discussed in Supplemental Section 8.2.2), the smoothed lidar and MAX-DOAS AODs are expected to capture the extinction between 0-2 km. The AERONET AODs would only be expected to be the same as the MAX-DOAS AODs if either a) there was no aerosol extinction above the boundary layer or b) the AERONET AODs were converted to a 0-4 km profile that was smoothed using the MAX-DOAS retrieval information. There was evidence of non-trivial aerosol extinction above 2 km and even sometimes above 4 km in the lidar measurements. An example of the variation in aerosol extinction at altitudes above 2km observed by the lidar was added to the supplemental (Fig. S9). An expanded discussion of how contributions of aerosol extinction above the boundary layer to the total AOD can be non-trivial was added on page 14 starting on line 5. This includes a reference to measurements of monthly average contribution of free tropospheric AOD to total AOD from satellite observations (Bourgeois et al., 2018, Atmos. Chem. Phys. DOI: https://www.atmos-chemphys.net/18/7709/2018/).

**3.** Page 9, line 3: According to Rodgers (2000), weighting function K should be equal to  $\delta y/\delta x$  and not  $\delta F/\delta x$ .

**Response: corrected on Page 9 line 8.**

**4.** Page 9: I think you should justify your choice of aerosol extinction and trace gas concentration a priori profiles. Did you perform sensitivity tests for the selection of these a priori profiles, especially in terms of scaling height ?

**Response**: Results and discussion of sensitivity analysis performed on the a-priori profiles was added to the Supplemental as Section 8.2. Sensitivity studies are shown for two case study days, Aug 23 and Sep 04, including results of a-priori scaling height of 0.3, 0.6 and 1.2 km. The new section is now referred to in the main manuscript on pages 9 line 22 and 10 line 2.

Regarding variations in the scale height (except from discussion in supplement 8.2): The a-priori with a scale height of 0.3 km decreased the AODs or VCD retrievals by 0-30% compared to the "base-case", depending on the day and species (Tables S18-S23). The retrievals using this a-priori resulted in greater chi-squared values for the h=0.3 profiles compared to the other profiles for some retrievals after 15:00 (Figs. S14-S16). This smaller a-priori scale height was not used for the retrievals because it was probably too restrictive in terms of the decrease in the a-priori values profile values between ~0.5-1 km under well-mixed, afternoon boundary layer conditions.

The a-priori with a scale height of 1.2 km increased the AOD or VCD retrieved values by ~20% compared to the "base-case", generally because there was little information content from the MAX-DOAS measurements above ~1.5 km and the retrieval reverted to the non-zero a-priori values (Figs. S21-S25). Consequently, this profile shape increased the proportion of aerosol extinction and trace-gas concentration present at >2 km altitude to the 0-4 km column value (see plot below and Figs. S21-S25). For some of the vertical profiles on both days, the retrieval reduced the aerosol extinction to zero by ~1 km but this a-priori resulted in non-zero values at 2-4 km (Figs. S21-S25). This larger scale height was not used for the retrievals because it resulted in non-zero values at high altitudes where there was little to no information content from the measurements (Figs. S7, S21-S25). Since the values at these higher altitudes were unknown and likely varied temporally, a scale height that allowed the values to be retrieved at zero was chosen (i.e., h= 0.6 km). An a-priori scale height that resulted in underestimated rather than overestimated retrievals of the pollution loading where there was a lack of information content was considered preferable.

**5.** Page 9, lines 30-31: To my knowledge the SCIATRAN RTM is not based on a Monte Carlo approach. This point should be clarified.

**Response: The reviewer is correct. "Monte Carlo" was removed on page 10 line 8.**

**6.** Page 10, line 10: The relative error of the a priori was set to 100% for the construction of the Sa matrix. Did you set the extra-diagonal terms to zero and, again did you perform sensitivity tests for the selection of this relative error value. Also related: nothing is said in the paper about the quality control of your MAX-DOAS retrievals. For instance, what are the typical degrees of freedom for signal (DOFS) values of your aerosol and trace gas MAX-DOAS retrievals and what is the level of agreement between measured DSCDs and those modelled using the retrieved profiles as input ?

**Response**: Extra-diagonal terms of  $S_a$  matrix were set to zero and this information was added on page 10 line 19. Sensitivity tests for this relative error value were not performed. Typical degrees of freedom of signal were 1.6-2.1 and 2.3-3.0 for the aerosol and trace-gas retrievals, respectively. This information was added to Supplemental Section 8. Statistical results of the linear correlations between the measured and modelled dSCDs of O4, SO2, and NO2 that includes a brief discussion of results were added to the supplemental (section 8.1 Tables S10 and S11). The presence of a "quality of retrieval" section the supplemental was added on page 8 lines 29-30 of the manuscript.

**7.** Comparisons between MAX-DOAS and AERONET AODs (Figs 4a-9a): MAX-DOAS significantly underestimates (by sometimes more than a factor of 2) AERONET AODs. What happens if you used a different a priori profile with a scaling height larger 0.6 km (e.g. 1.2 km)? Does it improve the agreement with AERONET without degrading the quality of your retrieval (see point 6/)? Also, is the application of a O4 scaling factor can improve the agreement with AERONET ?

**Response:** See response to reviewer comment #2 for discussion of why MAX-DOAS AODs are expected to be significantly less than the AERONET AODs during certain periods such as on Sep 04. As now shown in Supplemental section 8.2, increasing the scaling height of the MAX-DOAS AOD retrieval to 1.2 km from 0.6 km for Sep 04 increased the diurnal AOD by 14% (+/-13%) (Table S21). The AODs produced by the two scale heights were effectively equal within error for this day (Fig. S17). This result was expected since there appeared to be a significant contribution of aerosol extinction above the boundary layer height on this day that probably occurred on other days as well (see response to comment #2). The Aug 23 MAX-DOAS AODs using a scale height of 1.2 km were significantly greater than the h=0.6 km AODs between 15:30 and 16:45 but were significantly smaller after 17:00 with reduced degrees of freedom of signal (Fig. S14). The h=1.2 km AOD retrieval also produced a temporal trend less consistent with the lidar temporal trends in AOD compared to the h=0.6 km AODs (Fig. S14). Therefore, using an apriori with h=1.2 km for the AOD retrieval appeared to produce a lower quality retrieval for Aug 23. See response to reviewer comment #2 for why an O4 scaling factor was not applied.

**8.** Page 22, lines 1-3: You said that a major advantage of the MAX-DOAS technique is the simultaneous retrievals of total columns and vertical profiles of trace gases and aerosol extinctions without requiring a priori information. I am a bit puzzled by this sentence since a priori information is needed in the Optimal Estimation retrieval approach you used. A clarification is needed here.

Response: Agreed, this sentence was unclear. Sentence was modified Page 23 lines 1-3 to make it clear that the a priori information referred in this case to the aerosol characteristics that can be needed to retrieve accurate AODs from lidar compared to MAX-DOAS.

Technical corrections: '->' denotes 'should be replaced by'

1/'Honninger et al.' -> 'Hönninger et al.'

Response: Corrected throughout.

2/'Clemer et al.' -> 'Clémer et al.'

Response: Corrected throughout.

3/The first sentences of both Sections 2.1 and 2.2 are a bit redundant. I would start Section 2.1 by 'The MAX-DOAS instrument was operated at an elevation of  $\sim 10 \text{ m.}$  ...' and Section 2.2 by 'The MAX-DOAS instrument is a mini-DOAS spectrometer from Hoffmann Messtechnik GmbH measuring scattered sunlight. ....' or something similar.

Response: Updated based on reviewer comments on page 5 on lines 3 and 18.

4/Page 5, line 25: 5°C (C not in superscript)

Response: Corrected on page 5 line 26.

5/Page 22, line 1: 'total column' -> 'total columns'

Response: Corrected on page 23 line 6.

6/List of References: 'Atmospheric Meas. Tech.' -> 'Atmos. Meas. Tech.'; Same 'Atmospheric Chem. Phys.'

Response: This has been corrected throughout the list of references on pages 49 to 55.

7/Legends of Figures 4-9: You should add a short description of the error bars presented in the plots.

Response: A short description of the error bars was added to the last sentence in the captions of Figures 4-9.

**Validation of MAX-DOAS retrievals of aerosol extinction, SO2 and NO2 through comparison with lidar, sun photometer, Active-DOAS and aircraft measurements in the Athabasca Oil Sands Region.**

Zoë Y. W. Davis1, Udo Frieβ2, Kevin B. Strawbridge3, Monika Aggarwaal1, Sabour Baray4, Elijah G. Schnitzler5, Akshay Lobo44,6, Vitali E. Fioletov3, Ihab Abboud3, Chris A. McLinden3, Jim Whiteway1, Megan D. Willis5,7, Alex K. Y. Lee8, Jeff Brook3,9, Jason Olfert10, Jason O'Brien3, Ralf Staebler3, Hans D. Osthoff11, Cristian Mihele3, and Robert McLaren4.

1Department of Earth and Space Science, York University, Toronto, M3J 1P3, Canada

- 2Institute of Environmental Physics, Heidelberg, 69120, Germany
- 3 Environment and Climate Change Canada, Toronto, M3H 5T4, Canada
   4 Centre for Atmospheric Chemistry, York University, Toronto, M3J 1P3, Canada
   5 Department of Chemistry, University of Toronto, M5S 3H6, Canada
   6 now at Department of Orthopaedics, University of British Columbia, Vancouver, V5Z 1M9, Canada
   7 now at Chemical Sciences Division, Lawrence Berkeley National Lab, Berkeley, California, 94720, USA
- 8 Department of Civil and Environmental Engineering, National University of Singapore, 117576, Singapore 9 now at Dalla Lana School of Public Health, University of Toronto, M5S 3H6, Canada
  - 10 Department of Mechanical Engineering, University of Alberta, Edmonton, Alberta, T6G 1H9
  - 11 Department of Chemistry, University of Calgary, Calgary, T2N 1N4, Canada

20 Correspondence to: Zoë Davis (zoeywd@yorku.ca)

**Abstract.** Vertical profiles of aerosols,  $NO_2$ , and  $SO_2$  were retrieved from Multi-Axis Differential Optical Absorption Spectroscopy (MAX-DOAS) measurements at a field site in northern Alberta, Canada, during August and September 2013. The site is approximately 16 km north of two mining operations that are major sources of industrial pollution in the Athabasca Oil Sands Region. Pollution conditions during the study ranged from atmospheric background conditions to

- 25 heavily polluted with elevated plumes, according to the meteorology. This study aimed to evaluate the performance of the aerosol and trace gas retrievals through comparison with data from a suite of other instruments. Comparisons of AODs from MAX-DOAS aerosol retrievals, lidar vertical profiles of aerosol extinction, and AERONET sun photometer indicate good performance by the MAX-DOAS retrievals. These comparisons and modelling of the lidar S-ratio highlight the need for accurate knowledge of the temporal variation in the S-ratio when comparing MAX-DOAS and lidar data. Comparisons of
- 30 MAX-DOAS NO2 and SO2 retrievals to Pandora spectral sun photometer VCDs and Active-DOAS mixing ratios indicate good performance of the retrievals except when vertical profiles of pollutants within the boundary layer varied rapidly, temporally and spatially. Near-surface retrievals tended to overestimate Active-DOAS mixing ratios. The MAX-DOAS observed elevated pollution plumes not observed by the Active-DOAS, highlighting one of the instrument's main advantages. Aircraft measurements of SO2 were used to validate retrieved vertical profiles of SO2. Advantages of the MAX-
- 35 DOAS instrument include increasing sensitivity towards the surface and the ability to simultaneously retrieve vertical

profiles of aerosols and trace gases without requiring additional parameters such as the S-ratio. This complex dataset provided a rare opportunity to evaluate the performance of the MAX-DOAS retrievals under varying atmospheric conditions.

**1** Introduction**

- The Athabasca Oil sands operations in Alberta contain significant sources of industrial atmospheric pollutants such as sulphur dioxide (SO2) and nitrogen dioxide (NO2) (ECCC, 2018b, 2018c). Oil extraction and upgrading activities such as 5 Field Code Changed surface mining, acid gas flaring, and transporting materials in heavy hauler trucks emit aerosols and trace gas pollutants (Liggio et al., 2016). Pollutant emissions from the industrial smokestacks result in uplifted profiles with the potential to be transported farther downwind compared to emission released at the surface, particularly for stacks with high volume flow rates and temperatures that can rise high in the atmosphere (Zhang et al., 2018). While the Athabasca Oil Sands Region 10 (AOSR) experiences moderate annual average concentrations of SO2 relative to all Canadian in-situ stations, the short-term concentrations can be significantly higher than in most Canadian cities (Government of Canada, 2018). The AOSR contains **Field Code Changed** some of the few monitoring sites in Canada that experience peak 1-hour average concentrations of  $SO_2$  of greater than 70 ppb (Government of Canada, 2018), which is the new 2020 Canadian Ambient Air Quality Standard for SO2 (Canadian Field Code Changed Council of Ministers of the Environment, 2014). SO2 concentrations of up to 131 ppb were also observed by aircraft Field Code Changed measurements downwind of an AOSR industrial facility in 2013, approximately midway between Syncrude Mildred Lake 15 Plant and Fort McKay (Baray et al., 2018). High concentrations of SO2 over short durations are a health concern because Field Code Changed negative pulmonary and respiratory effects of inhalation can occur after exposure periods as small as 10 minutes (Health Field Code Changed Canada, 2016; WHO, 2006). Exposure to NO2 at high concentrations over short-term is also associated with significant health impacts (WHO, 2006) and NOx (NO + NO2) is a precursor to tropospheric ozone (O3), acid rain and fine particulate **Field Code Changed** 20 matter (Seinfeld and Pandis, 2006). Field Code Changed Emissions of  $NO_x$  and  $SO_2$  lead to the formation of nitrate and sulphate aerosols, which constitute a significant fraction of the  $PM_{2.5}$  air mass in urban and industrially-impacted regions (Pui et al., 2014). The highest peak and annual average  $PM_{2.5}$ Field Code Changed concentrations in Canada in 2016 were observed at two monitoring stations within Fort McMurray with annual averages of over 18 µg m-3 compared to 8 µg m-3 in an industrial area of Toronto, Ontario (Government of Canada, 2018). Exposure to **Field Code Changed** PM2.5 leads to adverse effects on respiratory and cardiovascular systems (WHO, 2006). 25 **Field Code Changed** In the troposphere, nearly all SO2 is oxidized to H2SO4 aerosol through reactions in the gas and aqueous phases. The hydroxyl (OH) radical initiates the oxidation route of SO2 in the gas phase, forming HOSO2 (Holloway and Wayne, 2010). Field Code Changed Sulfuric acid (H2SO4) is formed through further oxidation of HOSO2 
[revised manuscript text omitted]

|   | New Roman)                                                        |
|---|-------------------------------------------------------------------|
|   | Formatted: Font: (Default) +Headings (Tin
New Roman)           |
|   | Formatted: Font: (Default) +Headings (Tin
New Roman)           |
|   | Formatted: Font: (Default) +Headings (Tin
New Roman), Not Bold |
|   | Formatted: Font: (Default) +Headings (Tin
New Roman), Not Bold |
| Ì | Formatted: Superscript                                            |
|   | Formatted: Font: (Default) +Headings (Tin
New Roman)           |
|   | Formatted: Font: (Default) +Headings (Tin
New Roman)           |
|   | Field Code Changed                                                |
| Ì | Formatted: Not Highlight                                          |
| l | Formatted: Not Highlight                                          |
| Ì | Formatted: Not Highlight                                          |
| V | Formatted: Not Highlight                                          |
|   | Formatted: Not Highlight                                          |
| V | Formatted: Not Highlight                                          |
| V | Formatted: Not Highlight                                          |
| V | Formatted: Not Highlight                                          |
|   | Formatted: Not Highlight                                          |
|   | Formatted: Not Highlight                                          |
|   | Formatted: Not Highlight                                          |
| ١ | Formatted: Not Highlight                                          |
|   |                                                                   |

[revised manuscript text omitted]